# Laplacian Autoencoders for Learning Stochastic Representations

**Marco Miani**[1], **Frederik Warburg**[1],
**Pablo Moreno-Muñoz, Nicke Skafte Detlefsen, Søren Hauberg**
{mmia, frwa, pabmo, nsde, sohau}@dtu.dk
Technical University of Denmark

https://github.com/FrederikWarburg/LaplaceAE

## Abstract

Established methods for unsupervised representation learning such as variational autoencoders produce none or poorly calibrated uncertainty estimates making it difficult to evaluate if learned representations are stable and reliable. In this work, we present a Bayesian autoencoder for unsupervised representation learning, which is trained using a novel variational lower bound of the autoencoder evidence. This is maximized using Monte Carlo EM with a variational distribution that takes the shape of a Laplace approximation. We develop a new Hessian approximation that scales linearly with data size allowing us to model high-dimensional data. Empirically, we show that our Laplacian autoencoder estimates well-calibrated uncertainties in both latent and output space. We demonstrate that this results in improved performance across a multitude of downstream tasks.

## 1 Introduction

Unsupervised representation learning is a brittle matter. Consider the *classic autoencoder (*AE*)* (Rumelhart et al., 1986), which compresses data $x$ to a low-dimensional representation $z$ from which data is approximately reconstructed. The nonlinearity of the model implies that sometimes small changes to data $x$ give a large change in the latent representation $z$ (and sometimes not). Likewise, for some data, reconstructions are of low quality, while for others it is near perfect. Unfortunately, the model does not have a built-in quantification of its uncertainty, and we cannot easily answer when the representation is reliable and accurately reflects data.

The celebrated *variational autoencoder* (VAE) (Kingma and Welling, 2014; Rezende et al., 2014) address this concern directly through an explicit likelihood model $p(x|z)$ and a variational approximation of the representation posterior $p(z|x)$. Both these distributions have parameters predicted by neural networks that act similarly to the encoder–decoder pair of the classic autoencoder.

But is the VAE's quantification of reliability reliable? To investigate, we fit a VAE with a two-dimensional latent representation to the MNIST dataset (Lecun et al., 1998), and illustrate the predicted uncertainty of $p(x|z)$ in Fig. 1a. The model learns to assign high uncertainty to low-level image features such as edges but predicts its smallest values far away from the data distribution. Not only is such behavior counter-intuitive, but it is also suboptimal in terms of data likelihood (Sec. 1.1). Retrospectively, this should not be surprising as the uncertainty levels away from the data are governed by the extrapolatory behavior of the neural network determining $p(x|z)$. This suggests that perhaps uncertainty should be a derived quantity rather than a predicted one.

---

[1] Denotes equal contribution; author order determined by a simulated coin toss.

36th Conference on Neural Information Processing Systems (NeurIPS 2022).

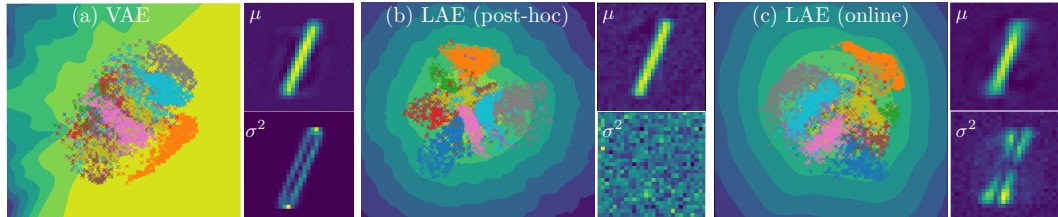

Figure 1: 2D latent representation of MNIST overlaid a heatmap that describes the decoder uncertainty (yellow/blue indicates a low/high variance of the reconstructions). To the right of the latent spaces, we show the mean and variance of a reconstructed image (yellow indicates high values). (a) The VAE learns to estimate high variance for low-level image features such as edges but fails at extrapolating uncertainties away from training data. (b) Applying post-hoc Laplace to the AE setup shows much better extrapolating capabilities, but fails in estimating calibrated uncertainties in output space. (c) Our online, sampling-based optimization of a Laplacian autoencoder (LAE) gives well-behaved uncertainties in both latent and output space.

From a Bayesian perspective, the natural solution is to form an (approximate) posterior over the weights of the neural networks. To investigate, we adapt a state-of-the-art implementation of a *post-hoc* Laplace approximation (Daxberger et al., 2021) of the weight posterior to the autoencoder domain. This amounts to training a regular autoencoder, and thereafter approximating the weight uncertainty with the Hessian of the loss (Sec. 1.1). Fig. 1b shows that uncertainty now grows, as intuitively expected, with the distance to the data distribution, but there seems to be little semantic structure in the uncertainty in output space. This suggests that while the post-hoc procedure is computationally attractive it is too simplistic.

**In this paper** we introduce a new framework for Bayesian autoencoders in unsupervised representation learning. Our method takes inspiration from the Laplace approximation to build a variational distribution on the neural network weights. We first propose a post-hoc LA for autoencoders; showcasing good out-of-distribution detection capabilities, but lack of properly calibrated uncertainties in-distribution. To address this, we develop a fast and memory-efficient Hessian approximation, which allows us to maximize a variational lower bound using Monte Carlo EM, such that model uncertainty is a key part of model training rather than estimated post-hoc. Fig. 1c gives an example of the corresponding uncertainty, which exhibits a natural and semantically meaningful behavior.

## 1.1 Background

**The VAE** is a latent variable model that parametrize the data density $p(\boldsymbol{x}) = \int p(\boldsymbol{x}|\boldsymbol{z})p(\boldsymbol{z})\mathrm{d}\boldsymbol{z}$ using a latent variable (representation) $\boldsymbol{z}$. Here $p(\boldsymbol{z})$ is a, usually standard normal, prior over the representation, and $p(\boldsymbol{x}|\boldsymbol{z})$ is a likelihood with parameters predicted by a neural network.

The nonlinearity of the likelihood parameters renders the marginalization of $\boldsymbol{z}$ intractable, and a variational lower bound of $p(\boldsymbol{x})$ is considered instead. To arrive at this, one first introduces a variational approximation $q(\boldsymbol{z}|\boldsymbol{x}) \approx p(\boldsymbol{z}|\boldsymbol{x})$ and write $p(\boldsymbol{x}) = \mathbb{E}_{q(\boldsymbol{z}|\boldsymbol{x})}\left[p(\boldsymbol{x}|\boldsymbol{z})^{p(\boldsymbol{z})}/q(\boldsymbol{z}|\boldsymbol{x})\right]$. A lower bound on $p(\boldsymbol{x})$ then follows by a direct application of Jensen's inequality,

$$\log p(\boldsymbol{x}) \geq \mathcal{L}_{\text{VAE}}(\boldsymbol{x}) = \mathbb{E}_{q(\boldsymbol{z}|\boldsymbol{x})}\left[\log p(\boldsymbol{x}|\boldsymbol{z})\right] - \text{KL}(q(\boldsymbol{z}|\boldsymbol{x})\|p(\boldsymbol{z})). \tag{1}$$

If we momentarily assume that $p(\boldsymbol{x}|\boldsymbol{z}) = \mathcal{N}(\boldsymbol{x}|\mu(\boldsymbol{z}), \sigma^2(\boldsymbol{z}))$, we see that optimally $\sigma^2(\boldsymbol{z})$ should be as large as possible away from training data in order to increase $p(\boldsymbol{x})$ on the training data (Appendix F). Yet this is not the observed empirical behavior in Fig. 1a. Since the $\sigma^2$ network is left untrained away from training data, its predictions depend on extrapolation. In practice, $\sigma^2$ takes fairly small values near training data (assuming the mean $\mu$ provides a reasonable data fit), and $\sigma^2$ extrapolates arbitrary even if this is suboptimal in terms of data likelihood. Similar remarks hold for other likelihood models $p(\boldsymbol{x}|\boldsymbol{z})$ and encoder distributions $q(\boldsymbol{z}|\boldsymbol{x})$: *relying on neural network extrapolation to predict uncertainty does not work*.

**The Laplace approximation** (Laplace, 1774; MacKay, 1992) is an integral part of our proposed solution. In the context of Bayesian neural networks, we seek the weight-posterior $p(\theta|\mathcal{D}) \propto \exp(-\mathcal{L}(\mathcal{D};\theta))$, where $\theta$ are network weights, $\mathcal{D}$ is the training data, and $\mathcal{L}$ is the applied loss

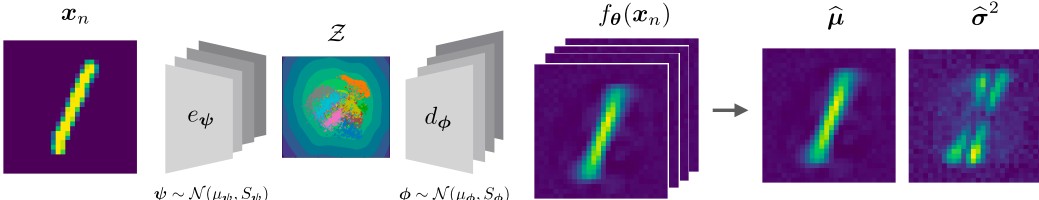

Figure 2: **Model overview.** We learn a distribution over parameters such that we can sample encoders $e_{\psi}$ and decoders $d_{\phi}$. This allow us to compute the empirical mean and variance in both the latent space $z$ and the output space $f_{\theta}(x_n) = d_{\phi}(e_{\psi}(x_n))$.

function interpreted as an unnormalized log-posterior. This is generally intractable and Laplace's approximation (LA) amounts to a second-order Taylor expansion around a chosen weight vector $\theta^*$

$$\log p(\boldsymbol{\theta}|\mathcal{D}) = \mathcal{L}^* + (\boldsymbol{\theta} - \boldsymbol{\theta}^*)^{\top}\nabla\mathcal{L}^* + \frac{1}{2}(\boldsymbol{\theta} - \boldsymbol{\theta}^*)^{\top}\nabla^2\mathcal{L}^*(\boldsymbol{\theta} - \boldsymbol{\theta}^*) + \mathcal{O}(\|\boldsymbol{\theta} - \boldsymbol{\theta}^*\|^3) \quad (2)$$

where we use the short-hand $\mathcal{L}^* = \mathcal{L}(\mathcal{D}; \theta^*)$. The approximation, thus, assumes that $p(\theta|\mathcal{D})$ is Gaussian. Note that when $\theta^*$ is a MAP estimate, the first order term vanishes and the second order term is negative semi-definite. We provide more details on the Laplace approximation in Appendix B. In practice, computing the full Hessian is too taxing both in terms of computation and memory, and various approximations are applied (Sec. 3).

## 2 Laplacian Autoencoders

We consider unsupervised representation learning from i.i.d. data $\mathcal{D} = \{x_n\}_{n=1}^{N}$ consisting of observations $x_n \in \mathbb{R}^D$. We also define a continuous latent space such that representations $z_n \in \mathbb{R}^K$. Similar to AEs (Hinton and Salakhutdinov, 2006), we consider two neural networks $e_{\psi} : \mathbb{R}^D \to \mathbb{R}^K$ and $d_{\phi} : \mathbb{R}^K \to \mathbb{R}^D$, widely known as the *encoder* and *decoder*. These have parameters $\boldsymbol{\theta} = \{\psi, \phi\}$. We refer to the composition of encoder and decoder as $f_{\boldsymbol{\theta}} = d_{\psi} \circ e_{\phi}$.

**Model overview.** The autoencoder network structure implies that we model the data as being distributed on a $K$-dimensional manifold parametrized by $\boldsymbol{\theta}$. We then seek the distribution of the *reconstruction* $x_{\text{rec}} = f_{\boldsymbol{\theta}}(x)$ given observation $x$, where the uncertainty comes from $\boldsymbol{\theta}$ being unknown,

$$p(x_{\text{rec}}|x, f) = \mathbb{E}_{\boldsymbol{\theta} \sim p(\boldsymbol{\theta}|x, f)}[p(x_{\text{rec}}|\boldsymbol{\theta}, x, f)]. \quad (3)$$

Notice that we explicitly condition on $f$, which is the operator $\boldsymbol{\theta} \mapsto f_{\boldsymbol{\theta}}$, even if this is not stochastic; this conditioning will become important later on to distinguish between the distribution deduced by $f$ and its linearization $f^{(t)}$. Mimicking the standard autoencoder reconstruction loss, we set $p(x_{\text{rec}}|\boldsymbol{\theta}, x, f) = \mathcal{N}(x_{\text{rec}}|f_{\boldsymbol{\theta}}(x), \mathbb{I})$. Since $p(\boldsymbol{\theta}|x, f)$ is unknown, the reconstruction likelihood Eq. (3) is intractable, and approximations are in order. Similar to Blundell et al. (2015), we resort to a Gaussian approximation, but rather than learning the variance variationally, we opt for LA. This will allow us to sample NNs and deduce uncertainties in both latent and output space as illustrated in Fig. 2.

**Intractable joint distribution.** Any meaningful approximate posterior over $\boldsymbol{\theta}$ should be similar to the marginal of the joint distribution $p(\boldsymbol{\theta}, x_{\text{rec}}|x, f)$. This marginal is

$$p(\boldsymbol{\theta}|x, f) = \mathbb{E}_{x_{\text{rec}} \sim p(x_{\text{rec}}|x, f)}[p(\boldsymbol{\theta}|x_{\text{rec}}, x, f)] \quad (4)$$

which can be bounded on a log-scale using Jensen's inequality,

$$\log p(\boldsymbol{\theta}|x, f) \geq \mathcal{L}_{\boldsymbol{\theta}} = \mathbb{E}_{x_{\text{rec}} \sim p(x_{\text{rec}}|x, f)}[\log p(\boldsymbol{\theta}|x_{\text{rec}}, x, f)]. \quad (5)$$

Our first approximation is a LA of $p(\boldsymbol{\theta}|x, f) \approx q^t(\boldsymbol{\theta}|x, f) = \mathcal{N}(\boldsymbol{\theta}|\boldsymbol{\theta}_t, \mathbf{H}_t^{-1})$, where we postpone the details on how to acquire $\boldsymbol{\theta}_t$ and $\mathbf{H}_t$. These will eventually be iteratively computed from the lower bound Eq. (5); hence the $t$ index. Fig. 3 (a) illustrates the situation thus far: $p(\boldsymbol{\theta}|x, f)$ is approximately Gaussian, but the non-linearity of $f$ gives $p(x_{\text{rec}}|x, f)$ a non-trivial density.

**Linearization for gradient updates.** Standard gradient-based learning can be viewed as a linearization of $f$ in $\boldsymbol{\theta}$, i.e. for a loss $\mathcal{L} = l(f_{\boldsymbol{\theta}})$, the gradient is $\nabla_{\boldsymbol{\theta}}\mathcal{L} = J_{\boldsymbol{\theta}}f_{\boldsymbol{\theta}}\nabla_f l(f)$, where $J_{\boldsymbol{\theta}}f_{\boldsymbol{\theta}}$ is the

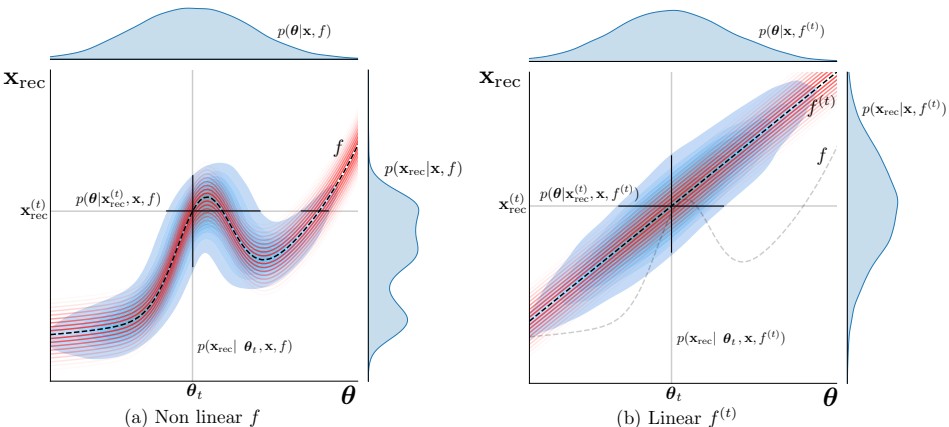

(a) Non linear $f$        (b) Linear $f^{(t)}$

Figure 3: Illustrative example for fixed $\boldsymbol{x}$. The likelihood for a fixed $\boldsymbol{\theta}_t$, shown by the columns, are assumed Gaussian $p(\boldsymbol{x}_{\text{rec}}|\boldsymbol{\theta}_t, \boldsymbol{x}, f) = N(f_{\boldsymbol{\theta}_t}(\boldsymbol{x}), \mathbb{I})$. We model the marginalised density $p(\boldsymbol{\theta}|\boldsymbol{x}, f)$ (first axis) over parameters $\boldsymbol{\theta}$ with Gaussians. With the additional assumption of linear $f$ then $p(\boldsymbol{x}_{\text{rec}}|\boldsymbol{x}, f)$ (second axis) is Gaussian. This makes the joint distribution tractable.

Jacobian of $f$. In a similar spirit, we linearize $f$ in $\boldsymbol{\theta}$ in order to arrive at a tractable approximation of $p(\boldsymbol{x}_{\text{rec}}|\boldsymbol{x}, f)$. Specifically, we perform a Taylor expansion around $\boldsymbol{\theta}_t$

$$f_{\boldsymbol{\theta}}(\boldsymbol{x}) = \underbrace{f_{\boldsymbol{\theta}_t}(\boldsymbol{x}) + J_{\boldsymbol{\theta}} f_{\boldsymbol{\theta}_t}(\boldsymbol{x})(\boldsymbol{\theta} - \boldsymbol{\theta}_t)}_{=: f_{\boldsymbol{\theta}}^{(t)}(\boldsymbol{x})} + \mathcal{O}\left(\|\boldsymbol{\theta} - \boldsymbol{\theta}_t\|^2\right) \tag{6}$$

where $f^{(t)}$ denote the associated first-order approximation of $f$. Under this approximation, the joint distribution $p(\boldsymbol{\theta}, \boldsymbol{x}_{\text{rec}}|\boldsymbol{x}, f)$ becomes Gaussian, $p(\boldsymbol{\theta}, \boldsymbol{x}_{\text{rec}}|\boldsymbol{x}, f^{(t)}) \approx \mathcal{N}(\boldsymbol{\theta}, \boldsymbol{x}_{\text{rec}}|\mu_t, \Sigma_t)$, with

$$\mu_t = \begin{pmatrix} \boldsymbol{\theta}_t \\ f_{\boldsymbol{\theta}_t}^{(t)}(\boldsymbol{x}) \end{pmatrix}, \text{ and } \Sigma_t = \begin{pmatrix} \mathbf{H}_t^{-1} & J_{\boldsymbol{\theta}} f_{\boldsymbol{\theta}_t}(\boldsymbol{x})^\top \\ J_{\boldsymbol{\theta}} f_{\boldsymbol{\theta}_t}(\boldsymbol{x}) & \left(J_{\boldsymbol{\theta}} f_{\boldsymbol{\theta}_t}(\boldsymbol{x})^\top \mathbf{H}_t J_{\boldsymbol{\theta}} f_{\boldsymbol{\theta}_t}(\boldsymbol{x})\right)^{-1} + \mathbb{I} \end{pmatrix}. \tag{7}$$

This approximation is illustrated in Fig. 3(b). We provide the proof in Appendix D.4.

**Iterative learning.** With these approximations we can readily develop an iterative learning scheme which updates $q(\boldsymbol{\theta}|\boldsymbol{x}, f)$. The mean of this approximate posterior can be updated according to a standard variational gradient step, $\boldsymbol{\theta}_{t+1} = \boldsymbol{\theta}_t + \lambda \nabla_{\boldsymbol{\theta}} \mathcal{L}_{\boldsymbol{x}_{\text{rec}}}$, where

$$\mathcal{L}_{\boldsymbol{x}_{\text{rec}}} = \mathbb{E}_{\boldsymbol{\theta} \sim q^t(\boldsymbol{\theta}|\boldsymbol{x}, f)}[\log p(\boldsymbol{x}_{\text{rec}}|\boldsymbol{\theta}, \boldsymbol{x}, f^{(t)})], \tag{8}$$

is a lower bound on Eq. (3), which we evaluate with a single Monte Carlo sample. Following the LA, the covariance of $q^{t+1}(\boldsymbol{\theta}|\boldsymbol{x}, f)$, should be the inverse of the Hessian of $\log p(\boldsymbol{\theta}|\boldsymbol{x}, f)$ at $\boldsymbol{\theta}_{t+1}$. Since this is intractable, we instead compute the Hessian of the lower bound Eq. (5)

$$\mathbf{H}_{t+1} = -\nabla_{\boldsymbol{\theta}}^2 \mathcal{L}_{\boldsymbol{\theta}}\big|_{\boldsymbol{\theta}_{t+1}} = \mathbb{E}_{p(\boldsymbol{x}_{\text{rec}}|\boldsymbol{x}, f^{(t)})}\left[-\nabla_{\boldsymbol{\theta}}^2 \log p(\boldsymbol{x}_{\text{rec}}|\boldsymbol{\theta}, \boldsymbol{x}, f^{(t)}) - \nabla_{\boldsymbol{\theta}}^2 \log q^t(\boldsymbol{\theta}|\boldsymbol{x}, f)\right]\Big|_{\boldsymbol{\theta}_{t+1}} \tag{9}$$

The last term can be approximated since $\nabla_{\boldsymbol{\theta}}^2 \log q^t(\boldsymbol{\theta}|\boldsymbol{x}, f)|_{\boldsymbol{\theta}=\boldsymbol{\theta}_{t+1}} = -\mathbf{H}_t + \mathcal{O}\left(\|\boldsymbol{\theta}_{t+1} - \boldsymbol{\theta}_t\|\right)$.

To efficiently cope with the $\mathcal{O}$-term, we introduce a parameter $\alpha$, such that the final approximation is

$$\mathbf{H}_{t+1} \approx (1-\alpha)\mathbf{H}_t + \mathbb{E}_{p(\boldsymbol{x}_{\text{rec}}|\boldsymbol{x}, f)}\left[-\nabla_{\boldsymbol{\theta}}^2 \log p\left(\boldsymbol{x}_{\text{rec}}|\boldsymbol{\theta}, \boldsymbol{x}, f^{(t)}\right)\right]\Big|_{\boldsymbol{\theta}_{t+1}} \tag{10}$$

$$= (1-\alpha)\mathbf{H}_t - J_{\boldsymbol{\theta}} f_{\boldsymbol{\theta}}^{(t)}{}^\top \nabla_{\boldsymbol{x}_{\text{rec}}}^2 \log p\left(\boldsymbol{x}_{\text{rec}}|\boldsymbol{\theta}_{t+1}, \boldsymbol{x}, f^{(t)}\right) J_{\boldsymbol{\theta}} f_{\boldsymbol{\theta}}^{(t)}. \tag{11}$$

where $J_{\boldsymbol{\theta}} f_{\boldsymbol{\theta}}^{(t)}$ is independent of which $\theta$ we evaluate in and $\nabla_{\boldsymbol{x}_{\text{rec}}}^2 \log p\left(\boldsymbol{x}_{\text{rec}}|\boldsymbol{\theta}_{t+1}, \boldsymbol{x}, f^{(t)}\right)$ is trivial to compute for common losses, i.e. for MSE it is the identity. The parameter $\alpha$ can be viewed as a geometric running average that is useful for smoothing out results computed on a minibatch instead of on the full training set, similar to momentum-like training procedures. It further allows for non-monotonically-increasing precision. Note that we revisit data during training, and the precision

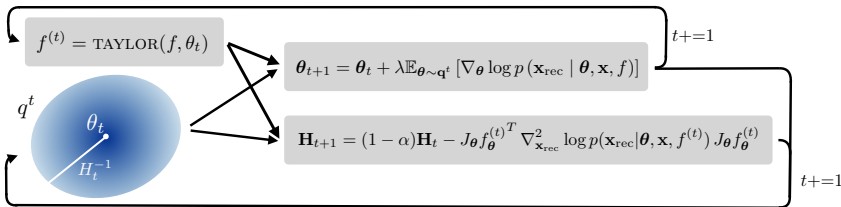

Figure 4: **Iterative training procedure.** Given a distribution $\mathbf{q}^t$ over parameters, and a linearized function $f^{(t)}$, compute first and second-order derivatives to update the distribution on parameters.

matrix is updated for every revisit. Thus, the forgetting induced by $\alpha$ is a wanted behavior to avoid infinite precision in the limit of infinite training time. In practice, we set $\alpha$ equal to the learning rate, where the practical intuition is that when $\alpha$ is small, the network uncertainty decreases faster.

The overall training procedure is summarized in Fig. 4. We initialize $q^0$ as a Gaussian with $\boldsymbol{\theta}_0 = 0$ and $\mathbf{H}_0 = \mathbb{I}$. We provide more details on the model, the linearization, and iterative learning in Appendix D.

**Why not just…?** The proposed training procedure may at first appear non-trivial, and it is reasonable to wonder if existing methods could be applied to similar results. Variational inference often achieves similar results to Laplace approximations, so could we use 'Bayes by Backprop' (Blundell et al., 2015) to get an alternative Gaussian approximate posterior over $\boldsymbol{\theta}$? Similar to the supervised experiences of Jospin et al. (2020), we, unfortunately, found this approach too brittle to allow for practical model fitting. But then perhaps a post-hoc LA as proposed by Daxberger et al. (2021) for supervised learning? Empirically, we found it to be important to center the approximate posterior around a point where the Hessian provides useful uncertainty estimates. Our online training moves in this direction as the Hessian is part of the procedure, but this is not true for the post-hoc LA.

Conceptually, we argue that our approach, while novel, is not entirely separate from existing methods. Our reliance on lower bounds makes the method an instance of variational inference (Jordan et al., 1999; Opper and Archambeau, 2009b), and we maximize the bounds using Monte Carlo EM (Cappé, 2009). We rely on a LA as our choice of variational distribution, which has also been explored by Park et al. (2019). Finally, we note that our linearization trick Eq. (6) has great similarities to classic extended Kalman filtering (Gelb et al., 1974).

## 3 Scaling the Hessian to Large Images

The largest obstacle to apply LA in practice stems from the Hessian matrix. This matrix has a quadratic memory complexity in the number of network parameters, which very quickly exceeds the capabilities of available hardware. To counter this issue, several approximations have been proposed (Ritter et al., 2018; Botev et al., 2017; Martens and Grosse, 2015b) that improve the scaling w.r.t. to the number of parameters. The currently most efficient Hessian implementations (Dangel et al., 2020; Daxberger et al., 2021) builds on the generalized Gauss-Newton (GGN) approximation of the Hessian

$$\nabla^2_{\boldsymbol{\theta}^{(l)}} \mathcal{L}(f_{\boldsymbol{\theta}}(\boldsymbol{x})) \approx J_{\boldsymbol{\theta}^{(l)}} f_{\boldsymbol{\theta}}(\boldsymbol{x})^\top \cdot \nabla^2_{\boldsymbol{x}_{\mathrm{rec}}} \mathcal{L}(\boldsymbol{x}_{\mathrm{rec}}) \cdot J_{\boldsymbol{\theta}^{(l)}} f_{\boldsymbol{\theta}}(\boldsymbol{x}), \tag{12}$$

for a single layer $l$, which neglects second order derivatives of $f$ w.r.t. the parameters. Besides, the computational benefits of this approximations, previous works on LA (Daxberger et al., 2021) relies on GGN to ensure that the Hessian is always semi-negative definite. In contrast, the model presented in Sec. 2 implies that GGN is no longer a practical and unprincipled trick, but rather the exact Hessian (Immer et al., 2021b).

Albeit relying on first order derivates, the layer-block-diagonal GGN, which assumes that layers are independent of each other, scales quadratically with the *output* dimension of the considered neural network $f$. This lack of scaling is particularly detrimental for convolutional layers as these have low parameter counts, but potentially very high output dimensions.

Expanding $J_{\boldsymbol{\theta}^{(l)}} f_{\boldsymbol{\theta}}(\boldsymbol{x})$ with the chain rule, one realizes that the Jacobian can be computed as a function of the Jacobian of the next layer. Fig. 6 illustrate that an intermediate quantity $M$, which is initialised as $\nabla^2_{\boldsymbol{x}_{\mathrm{rec}}} \mathcal{L}(\boldsymbol{x}_{\mathrm{rec}})$, can be efficiently backpropagated through multiplication with the Jacobian

| APPROXIMATIONS | MEMORY | TIME |
|---|---|---|
| Block diag. | $\mathcal{O}(R_m^2 + W_s^2)$ | $\mathcal{O}(R_s^2 + W_s^2)$ |
| KFAC | $\mathcal{O}(R_s^2 + W_s)$ | $\mathcal{O}(R_s^2 + W_s)$ |
| Exact diag. | $\mathcal{O}(R_m^2 + W_s)$ | $\mathcal{O}(R_s^2 + W_s)$ |
| Approx. diag. (ours) | $\mathcal{O}(R_m + W_s)$ | $\mathcal{O}(R_s + W_s)$ |
| Mixed diag. (ours) | $\mathcal{O}(R_m + W_s)$ | $\mathcal{O}(R_s + W_s)$ |

Table 1: **Memory & time complexity of Hessian approximations**. For an $L$-layer network, let $R_m = \max_{l=0...L} |x^{(l)}|$, $R_s = \sum_l |x^{(l)}|$, $R_s^2 = \sum_l |x^{(l)}|^2$, and $W_s = \sum_l |\boldsymbol{\theta}^{(l)}|$. Only our approximation scales linearly with both the output resolution and parameters.

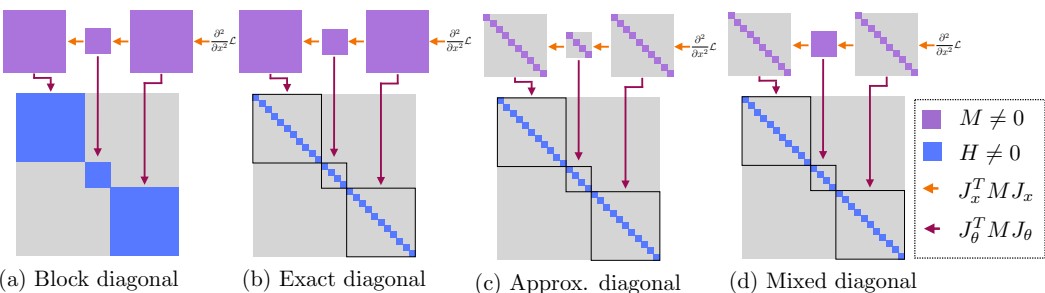

(a) Block diagonal  (b) Exact diagonal  (c) Approx. diagonal  (d) Mixed diagonal

Figure 5: **Comparison of Hessian approximation methods.** Common approximations (a–b) scale quadratically with the output resolution. Our proposed approximate and mixed diagonal Hessians (c–d) scale linearly with the resolution. This is essential for scaling the LAE to large images.

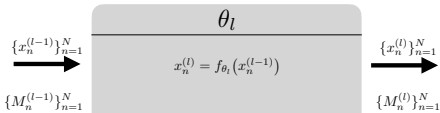

**diagonal** approximation of the Hessian as enerally too large to store and invert. To ts **exact diagonal** (Daxberger et al., 2021) ters, but still scales quadratically w.r.t. the

To scale our Laplacian autoencoder to high-dimensional data, we propose to approximate the diagonal of the Hessian rather than relying on exact computations. This is achieved by only backpropagating a diagonal form of $M$ as illustrated in Fig. 5(c). This assumes that features from the same layer are uncorrelated and consequently have linear complexity in both time and memory with respect to the output dimension (Tab. 1). This makes it viable for our model.

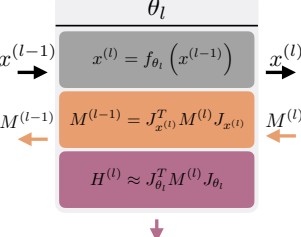

Figure 6: Forward pass of feature map $x$ for layer $l$ with parameters $\boldsymbol{\theta}^{(l)}$ and extended backward pass in which $M$ is backpropagated to previous layers. Via the chain rule and $M$ the Hessian of each layer can be computed efficiently.

We can further tailor this approximation to the autoencoder setting by leveraging the bottleneck architecture. We note that the quadratic scaling of the exact diagonal Hessian is less of an issue in the layers near the bottleneck than in the layers closer to the output space. We can therefore dynamically switch between our approximate diagonal and the exact one, depending on the feature dimension. This lessens the approximation error while remaining tractable in practice. We provide more details on the fast hessian computations in Appendix E.

## 4   Related Work

Deep generative models, and particularly the family of variational auto-encoders (VAEs) (Kingma and Welling, 2014; Rezende et al., 2014), address unsupervised representation learning from a probabilistic viewpoint by approximating the posterior over the representation space. Despite their widespread adoption, model parameters are still *deterministic* and sensitive to ill-suited local minima, e.g. over-fitted to training data (Zhang et al., 2021), which may cause poor generalization. The Bayesian NNs favour inference over the NN weights for addressing such issues (MacKay, 1995; Neal, 1996). This approach deduces distributions on data space by learning posterior distributions on the parameter space. However, several shortcomings (Wenzel et al., 2020), e.g. expensive training,

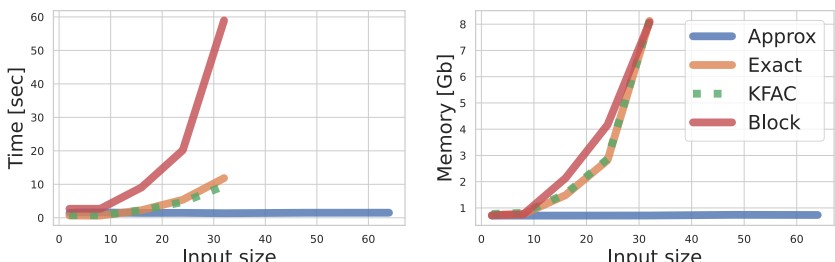

Figure 7: **Memory & time usage of Hessian approximations.** The exact and KFAC scales poorly with the image resolution. In contrast, our proposed approximate diagonal Hessian scales linearly.

tuning, and implementations, often limit their applicability to autoencoder-style models. Alternatively, other methods such as deep ensembles (Lakshminarayanan et al., 2017), stochastic weight averaging (SWAG) (Maddox et al., 2019) or Monte-Carlo dropout (Gal and Ghahramani, 2016) also promise Bayesian approximations to NN weight's posterior, but at the cost of increased training time, poor empirical performance or limited Bayesian interpretation.

As demonstrated by Daxberger et al. (2021) LA is a scalable and well-behaved alternative to Bayesian NNs if used *post-hoc* to approximate the intractable posterior over the weights after *maximum-a-posteriori* (MAP) training in classification and regression. The general utility of LA has also motivated its use as an approximation to the marginal likelihood over NN weights. Recent methods, including Daxberger et al. (2021), have explored this path to find model hyperparameters (Immer et al., 2021a) or learning invariances (Immer et al., 2022). However, its computational burden, for instance in Hessian matrices, has prescribed diagonal or Kronecker factored approximations (Ritter et al., 2018; Martens and Grosse, 2015a; Botev et al., 2017), which are now widely used for second-order optimization. We provide more details on the connections to existing hessian based methods Daxberger et al. (2021); Zhang et al. (2017); Khan et al. (2017), Bayes by Backpropagation Blundell et al. (2015) and Adam Kingma and Ba (2015a) in Appendix C.

The *full* Bayesian perspective on VAE weights was first explored by Daxberger and Hernández-Lobato (2019) which we find similar in spirit to our work. In contrast to them (1) we follow a principled Bayesian derivation. (2) Neither do we depend on Hamiltonian Monte-Carlo sampling, which is generally hard to scale to efficient training.

## 5   Experiments

First, we demonstrate the computational advantages of the proposed Hessian approximation, and then that our sampling-based training leads to well-calibrated uncertainties that can be used for OOD detection, data imputation, and semi-supervised learning. For all the downstream tasks we consider the following baselines: AE (Hinton and Salakhutdinov, 2006) with constant and learned variance, VAE (Rezende et al., 2014; Kingma and Welling, 2014), Monte-Carlo dropout AE (Gal and Ghahramani, 2016) and Ensembles of AE (Lakshminarayanan et al., 2017). We extend StochManDetlefsen et al. (2021) with the Hessian backpropagation for the approximate and mixed diagonals. The training code is implemented in PyTorch and available[2]. Appendix A provides more details on the experimental setup.

**Efficient Hessian Approximation.** For practical applications, training time and memory usage of the Hessian approximation must be kept low. We here show that the proposed approximate diagonal Hessian is sufficient and even outperforms other approximations when combined with our online training.

Fig. 7 show the time and memory requirement for different approximation methods as a function of input size for a 5-layer convolutional network that preserves channel and input dimension. As baselines we use efficient implementations of the exact and KFAC approximation (Daxberger et al., 2021; Dangel et al., 2020). The exact diagonal approximation run out of memory for an $\sim 36 \times 36 \times 3$ image on a 11 Gb NVIDIA GeForce GTX 1080 Ti. In contrast, our approximate diagonal Hessian scales linearly with the resolution, which is especially beneficial for convolutional layers.

---

2 https://github.com/FrederikWarburg/LaplaceAE

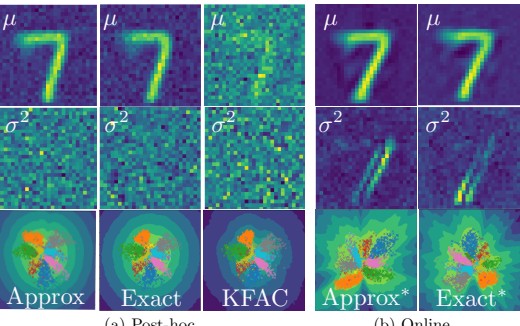

| Hessian | $-\log p(x)\downarrow$ | MSE $\downarrow$ |
|---|---|---|
| KFAC | $9683.9 \pm 2455.0$ | $121.6 \pm 24.5$ |
| Exact | $283.3 \pm 88.6$ | $27.1 \pm 0.9$ |
| Approx | $232.0 \pm 65.5$ | $26.6 \pm 0.6$ |
| Exact* | $25.8 \pm 0.2$ | $25.7 \pm 0.2$ |
| Approx* | $25.9 \pm 0.4$ | $25.8 \pm 0.4$ |

(a) Post-hoc  (b) Online

Table 2: Online training (indicated by *) outperforms post-hoc LA. The approximate diagonal has similar performance to the exact diagonal for both post-hoc and online LA.

Figure 8: Mean and variance of 100 sampled NN.

Tab. 2 shows that the exact or approximate Hessian diagonal has similar performance for both post-hoc and online training. Using post-hoc LA results in good mean reconstructions (low MSE), but each sampled NN does not give good reconstructions (low $\log p(x)$). Using our online training procedure results in a much higher log-likelihood. This indicates that every sampled NN predicts good reconstructions.

Fig. 8 shows the latent representation, mean, and variance of the reconstructions with the KFAC, exact and approximate diagonal for both post-hoc and online setup. Note that the online training makes the uncertainties better fitted, both in latent and data space. These well-fitted uncertainties have several practical downstream applications, which we demonstrate next.

**Out-of-Distribution (OOD) Detection** capabilities are critical for identifying distributional shifts, outliers, and irregular user inputs, which can hinder the propagation of erroneous decisions in an automated system. We evaluate OOD performance on the commonly used benchmarks (Nalisnick et al., 2019b), where we use FASHIONMNIST (Xiao et al., 2017) as in-distribution and MNIST (Lecun et al., 1998) as OOD. Fig. 9 (c) shows that our online LAE outperforms existing models in both log-likelihood and Typicality score (Nalisnick et al., 2019a). This stems from the calibrated model uncertainties, which are exemplified in the models ability to detect OOD examples from the uncertainty deduced in latent and output space; see Fig. 9 (a,b) for ROC curves.

Fig. 10 shows distribution of the output variances for in- and OOD data. This illustrates that using LA improves OoD detection. Furthermore, the online training improves the model calibration.

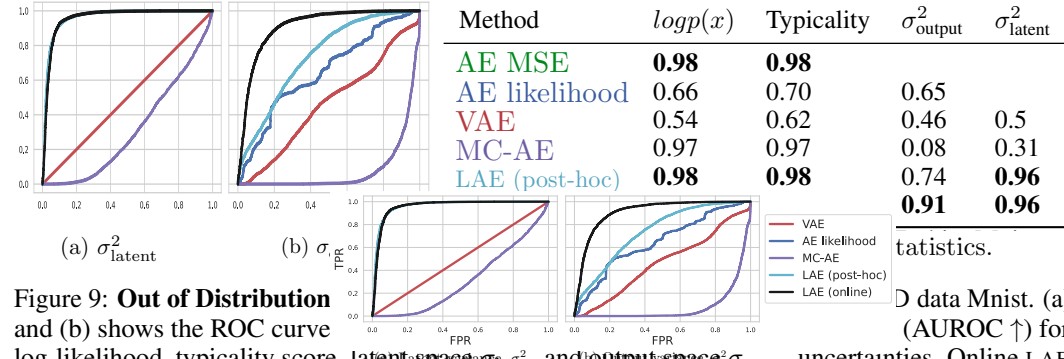

| Method | $logp(x)$ | Typicality | $\sigma^2_{output}$ | $\sigma^2_{latent}$ |
|---|---|---|---|---|
| AE MSE | **0.98** | **0.98** | | |
| AE likelihood | 0.66 | 0.70 | 0.65 | |
| VAE | 0.54 | 0.62 | 0.46 | 0.5 |
| MC-AE | 0.97 | 0.97 | 0.08 | 0.31 |
| LAE (post-hoc) | **0.98** | **0.98** | 0.74 | **0.96** |
| | | | **0.91** | **0.96** |

(a) $\sigma^2_{latent}$  (b) $\sigma_r$

Figure 9: **Out of Distribution** ... and (b) shows the ROC curve ... OOD data Mnist. (a) (AUROC ↑) for log-likelihood, typicality score, latent space $\sigma^2_{latent}$ and output space $\sigma^2_{output}$ uncertainties. Online LAE is able to discriminate between in and OOD using the deduced variances in latent and output space.

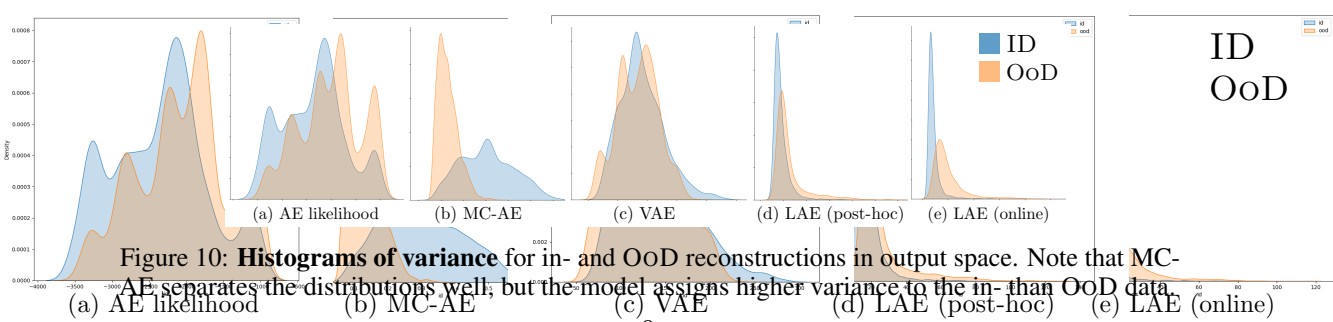

(a) AE likelihood  (b) MC-AE  (c) VAE  (d) LAE (post-hoc)  (e) LAE (online)

Figure 10: **Histograms of variance** for in- and OOD reconstructions in output space. Note that MC-AE separates the distributions well, but the model assigns higher variance to the in- than OoD data.

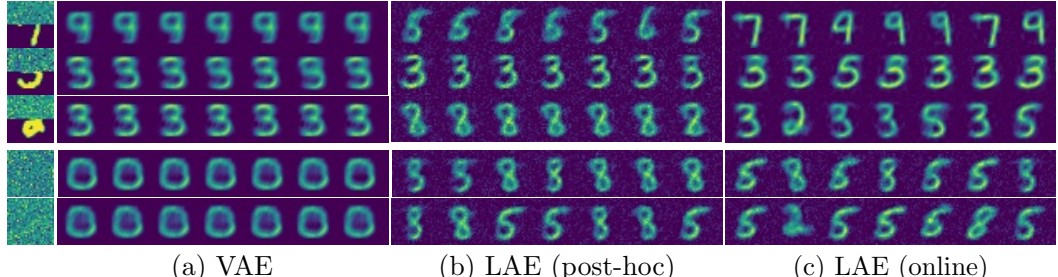

| (a) VAE | (b) LAE (post-hoc) | (c) LAE (online) |

Figure 11: **Missing data imputation & generative capabilities.** Online training of LAE improves representational robustness. This is exemplified by the multimodal behavior in the data imputation (top rows) that accurately model the ambiguity in the data. The bottom two rows show that the LAE is able to generate crisp digits from random noise.

| Method | MSE ↓ | $\log p(x)$ ↑ | Acc. ↑ | ECE ↓ | MCE ↓ | RMSCE ↓ |
|---|---|---|---|---|---|---|
| Classifier | | | **0.53** | 0.16 | 0.25 | 0.18 |
| VAE | 104.73 | -104.75 | 0.22 | 0.18 | 0.34 | 0.19 |
| MC-AE | 106.05 | -106.05 | 0.45 | 0.28 | 0.38 | 0.29 |
| AE Ensemble | **96.23** | **-100.94** | **0.53** | **0.12** | 0.2 | **0.13** |
| LAE (post-hoc) | 101.62 | -107.25 | 0.51 | 0.16 | 0.27 | 0.18 |
| LAE (online) | 99.59 | -106.29 | **0.53** | **0.12** | **0.16** | **0.13** |

Table 3: Reconstruction quality measured by the MSE and log-likelihood for the data imputation. Our well-calibrated uncertainties propagates to the MNIST classifier and improves the calibration metrics ECE, MCE and RMSCE.

**Missing Data imputation.** Another application of stochastic representation learning is to provide distributions over unobserved values (Rezende et al., 2014). In many application domains, sensor readings go missing, which we may mimic by letting parts of an image be unobserved. Rezende et al. (2014) show that we can then draw samples from the distribution of the entire image conditioned on the observed part, by imputing the missing pixels with noise and repeatedly encode and decode while keeping observed pixels fixed. Fig. 11 show samples using this procedure from a VAE, a post-hoc LAE and our online LAE, where we only observe the lower half of an MNIST image. This implies ambiguity about the original digit, e.g. the lower half of a "5" could be a "3" and similarly a "7" could be a "9". Our LAE captures this ambiguity, which is exemplified by the multi-modal reconstructions from the sampled networks in Fig. 11. The baselines only capture unimodal reconstructions.

Capturing the ambiguity of partly missing data can improve downstream tasks such as the calibration of an image classifier. In Fig. 11 (c) we demonstrate how averaging the predictions of a simple classifier across reconstructions improves standard calibration metrics. This is because the classifier inherits the uncertainty and ambiguity in the learned representations. A deep ensemble of AEs performs similarly to ours, but comes at the cost of training and storing multiple models.

When the entire input image is missing, the imputation procedure can be seen as a sampling mechanism, such that our LAE can be viewed as a generative model. The bottom rows in Fig. 11 show that the LAE indeed does generate sharp images from a multi-modal distribution.

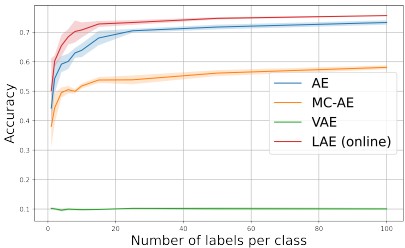

Figure 12: Accuracy as an function of the number of labels per class on MNIST.

| Attribute | AE | VAE | MC-AE | LAE* |
|---|---|---|---|---|
| Arched Eyebrows | 0.50 | 0.52 | 0.55 | 0.60 |
| Attractive | 0.52 | 0.50 | 0.49 | 0.53 |
| Bald | 0.98 | 0.98 | 0.98 | 0.98 |
| Wearing Lipstick | 0.52 | 0.49 | 0.50 | 0.54 |
| Heavy Makeup | 0.45 | 0.52 | 0.49 | 0.56 |
| Overall | 0.73 | 0.72 | 0.73 | **0.74** |

Table 4: Semi-supervised classification accuracy on CELEBA using only 10 labeled datapoints. * refers to online LAE.

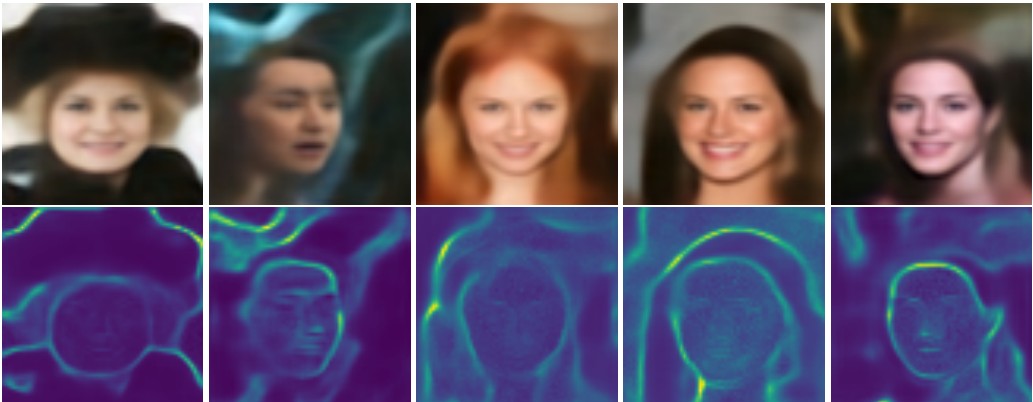

Figure 13: **Sample reconstructions on CELEBA.** The top row shows the mean reconstruction and the bottom row shows the variance of the reconstructed images.

**Semi-supervised learning** combines a small amount of label data with a large amount of unlabeled data. The hope is that the structure in the unlabeled data can be used to infer properties of the data that cannot be extracted from a few labeled points. Embedding the same labeled data point multiple times using a stochastic representation scales up the amount of labeled data that is available during training.

Fig. 12 shows the accuracy of a $K$-nearest neighbor classifier trained on different amounts of labeled data from the MNIST dataset. For all models with a stochastic encoder, we encode each labeled datapoint $100$ times and repeat the experiment $5$ times. When only a few labels per class are available (1-20) we clearly observe that our LAE model outperforms all other models, stochastic and deterministic. Increasing the number of labels beyond $100$ per class makes the AE and LAE equal in their classification performance with the AE model eventually outperforming the LAE model.

In Tab. 4 we conduct a similar experiment on the CELEBA (Liu et al., 2015) facial dataset, where the the task is to predict $40$ different binary labels per data point. When evaluating the overall accuracy of predicting all $40$ facial attributes, we see no significant difference in performance. However, when we zoom in on specific facial attributes we gain a clear performance advantage over other models. Fig. 13 shows the mean and variance of five reconstructed images. The online LAE produces well-calibrated uncertainties in the output space and scales to large images.

**Limitations.** Empirically, the LAE improvements are more significant for overparameterized networks. The additional capacity seems to help the optimizer find a local mode where a Gaussian fit is appropriate. It seems the regularization induced by marginalizing $\theta$ compensates for the added flexibility.

# 6 Conclusion

In this paper, we have introduced a Bayesian autoencoder that is realized using Laplace approximations. Unlike current models, this Laplacian autoencoder produces well-behaved uncertainties in both latent and data space. We have proposed a novel variational lower-bound of the autoencoder evidence and an efficient way to compute its Hessian on high dimensional data that scales linearly with data size. Empirically, we demonstrate that our proposed model predicts reliable stochastic representations that are useful for a multitude of downstream tasks: out-of-distribution detection, missing data imputation, and semi-supervised classification. Our work opens the way for fully Bayesian representation learning where we can marginalize the representation in downstream tasks. We find this to consistently improve performance.

**Acknowledgments and Disclosure of Funding**

This work was supported by research grants (15334, 42062) from VILLUM FONDEN. This project has also received funding from the European Research Council (ERC) under the European Union's Horizon 2020 research and innovation programme (grant agreement 757360). This work was funded in part by the Novo Nordisk Foundation through the Center for Basic Machine Learning Research in Life Science (NNF20OC0062606) and by the Pioneer Centre for AI, DNRF grant number P1.

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
