# Laplacian Autoencoders for Learning Stochastic Representations

**Marco Miani**[1], **Frederik Warburg**[1],
**Pablo Moreno-Muñoz, Nicke Skafte Detlefsen, Søren Hauberg**
{mmia, frwa, pabmo, nsde, sohau}@dtu.dk
Technical University of Denmark

https://github.com/FrederikWarburg/LaplaceAE

## — Supplementary Material —

The supplementary material is organized as follows. First, we give more technical details on the experiments. Second, we discuss the Laplace approximation and the "mean shift" issue encountered in non-local maxima. Third, we elaborate more on the difference between related works and our proposed method. Fourth, we present a much more thorough explanation of the proposed method accompanied by relevant proofs. Fifth, we present more details on the hessian derivations.

## A   Experimental details

All experiments was conducted on one of the following datasets: MNIST (Lecun et al., 1998), FASHIONMNIST (Xiao et al., 2017) and CELEBA (Liu et al., 2015). For training and testing the default splits were used. For validation, we sampled 5000 data points randomly from the training sets. All images were normalized to be in the $[0, 1]$ range and for the CELEBA dataset the images were resized to $64 \times 64$ additionally.

If nothing else is stated, the models were trained with the following configuration: we used Adam optimizer (Kingma and Ba, 2015b) with a learning rate of 0.001 and default Pytorch settings (Paszke et al., 2019). The learning rate was adjusted using *ReduceLROnPlateau* learning rate scheduler with parameters *factor=0.5* and *patience=5*, meaning that the learning rate was halved whenever 5 epochs had passed where the validation loss had not decreased. The mean squared error loss was used as the reconstruction loss in all models. Models were trained until convergence, defined as whenever the validation loss had not decreased for 8 epochs. Models trained on MNIST and FASHIONMNIST used a batch size of 64 and on CELEBA a batch size of 128 was used.

Model-specific details:

- VAE: Models were trained with KL-scaling of 0.001. We use two encoders and two decoders, such that the model has twice the number of parameters compared to the other models.
- MC-AE: Models were trained with dropout between all trainable layers with probability $p = 0.2$. We keep the same dropout rate during testing.
- EMSEMBLE-AE: Each ensemble consists of 5 models, each initialized with a different random seed.
- LAE (POSTHOC): For experiments with linear layers we used the Laplace Redux (Daxberger et al., 2021) implementation. For convolutions, we found it necessary to use our proposed hessian approximation. We use a diagonal approximation of the hessian in all experiments. After fitting the hessian, we optimize for the prior precision using the marginal likelihood (Daxberger et al., 2021). We use 100 MC sampling in all experiments.

---

[1] Denotes equal contribution; author order determined by a simulated coin toss.

36th Conference on Neural Information Processing Systems (NeurIPS 2022).

- LAE (ONLINE): We use the exact diagonal in experiments with linear layers and the mixed diagonal approximation in all experiments with convolutional layers. We use a hessian memory factor of $0.0001$ and sample only 1 network per iteration. We found that it was not necessary to optimize for the prior precision when trained online.

## A.1 Hessian approximation

We use a linear encoder-decoder with three layers in the encoder and decoder, TANH activation functions, and latent size 2. We choose this architecture as Laplace Redux (Daxberger et al., 2021) supports various hessian approximations for this simple network. We use Laplace Redux for all post-hoc experiments except for the approximate diagonal hessian.

## A.2 Out-of-distribution

We use a convolutional encoder-decoder architecture. The encoder consisted of a CONV2D, TANH, MAXPOOL2D, CONV2D, TANH, MAXPOOL2D, LINEAR, TANH, LINEAR, TANH, LINEAR, where the decoder was mirrored but with nearest neighbour Upsampling rather than MAXPOOL2D. We used a latent size of 2 in these experiments for all models.

## A.3 Missing data imputation

To elaborate on the procedure, we reconstruct 5 samples from the half/fully masked image. For each of these reconstructions, we make 5 more reconstructions and take the average of these reconstructions. The intuition is that the first stage explores the multi-modal behavior of the reconstructions. In the second stage, the uncertainty of the reconstructed digit is reduced, and each sample will reconstruct the same modality. By averaging over these modalities, we achieve a more crisp reconstruction. We use the same architecture as in the hessian approximation experiment.

## A.4 Semi-supervised learning

For the experiments on MNIST, we use a single convergence checkpoint for each model. We use the same model architecture as in the hessian approximation. We did 5 repetitions for each model where we first sampled $n$ labels from each of the 10 classes from the validation set, then embedded the $10 \times n$ data points into latent space, trained a KNN classifier on the embedded points and finally evaluated the accuracy of the classifier on the remaining validation set. This procedure was repeated for different values of $n$ in the $[1, 100]$ range. For the stochastic encoders (VAE, MC-AE, LAE), we repeated the embedding step 100 times with the goal that the uncertainty could help the downstream classification. For the KNN classifier we use cross-validation ($K = 2$) to find the optimal number of nearest neighbors.

For the experiments on CELEBA we repeated the exact same experiments but with a fixed value if $n = 10$. Additionally, the classifier was changed to a multi-label version KNN-classifier to accommodate the multiple binary features in the dataset. For CELEBA we use a convolutional architecture. The encoder consists of 5 convolutional layers with TANH and MAXPOOL2D in between each parametric layer. We use a latent size of 64. The decoder mirrors the encoder, but we replace MAXPOOL2D with nearest neighbor upsampling.

# B  Laplace Approximation

Laplace approximation is an operator that maps local properties (derivatives) to global properties. The idea is to infer a density on every point based on the curvature in a single point. This is done through a Taylor expansion.

Given a vectorial space $\Theta$ of size $D$, let $P(\boldsymbol{\theta})$ be an arbitrary distribution on $\Theta$ and let $\boldsymbol{\theta}^* \in \Theta$ be an arbitrary point. Consider the second order Taylor expansion of the log density around $\boldsymbol{\theta}^*$

$$\ln P(\boldsymbol{\theta}) = \ln P(\boldsymbol{\theta}^*) + \nabla_{\boldsymbol{\theta}} \ln P(\boldsymbol{\theta}^*)(\boldsymbol{\theta} - \boldsymbol{\theta}^*) + \frac{1}{2}(\boldsymbol{\theta} - \boldsymbol{\theta}^*)^\top \nabla_{\boldsymbol{\theta}}^2 \ln P(\boldsymbol{\theta}^*)(\boldsymbol{\theta} - \boldsymbol{\theta}^*) + \mathcal{O}(\|\boldsymbol{\theta} - \boldsymbol{\theta}^*\|^3),$$

where

$$[\nabla_{\boldsymbol{\theta}} \ln P(\boldsymbol{\theta}^*)]_i = \frac{\partial}{\partial \boldsymbol{\theta}_i} \ln P(\boldsymbol{\theta})\Big|_{\boldsymbol{\theta}=\boldsymbol{\theta}^*} \qquad [\nabla_{\boldsymbol{\theta}}^2 \ln P(\boldsymbol{\theta}^*)]_{ij} = \frac{\partial^2}{\partial \boldsymbol{\theta}_i \partial \boldsymbol{\theta}_j} \ln P(\boldsymbol{\theta})\Big|_{\boldsymbol{\theta}=\boldsymbol{\theta}^*}.$$

are the first and second-order derivatives.

Note that if $P(\boldsymbol{\theta})$ is a Gaussian, then its log density is a second order polynomial and the second order Taylor expansion is exact. This implies that starting from a Gaussian, the Laplace approximation can infer the full exact density just from the values of $\nabla_{\boldsymbol{\theta}} \ln P$ and $\nabla_{\boldsymbol{\theta}}^2 \ln P$ in a single point $\boldsymbol{\theta}^*$.

The main takeaway from the Laplace approximation is the strong, intrinsic, tie between the covariance matrix and the negative inverse of the hessian of the log probability: $-(\nabla_{\boldsymbol{\theta}}^2 \ln P(\boldsymbol{\theta}^*))^{-1}$.

## B.1 If $\boldsymbol{\theta}^*$ is a local maxima

If $\nabla_{\boldsymbol{\theta}} \ln P(\boldsymbol{\theta}^*) = 0$ the Taylor expansion consists of only two terms and the Laplace derivation is easier. We also present this in order to develop intuition, although this case is a subcase of the non-local maxima case. In the next section, we will consider the more general setting.

Define the Gaussian

$$\text{LAPLACE}_{\max}(\boldsymbol{\theta}^*; P) := \mathcal{N}\left(\boldsymbol{\theta}|\mu = \boldsymbol{\theta}^*, \sigma^2 = -(\nabla_{\boldsymbol{\theta}}^2 \ln P(\boldsymbol{\theta}^*))^{-1}\right), \tag{1}$$

which has density

$$Q(\boldsymbol{\theta}) := \frac{P(\boldsymbol{\theta}^*)}{Z_Q} e^{\frac{1}{2}(\boldsymbol{\theta}-\boldsymbol{\theta}^*)^\top (\nabla_{\boldsymbol{\theta}}^2 \ln P(\boldsymbol{\theta}^*))^{-1}(\boldsymbol{\theta}-\boldsymbol{\theta}^*)},$$

where $Z_Q = P(\boldsymbol{\theta}^*)\sqrt{(-2\pi)^D \det(\nabla_{\boldsymbol{\theta}}^2 \ln P(\boldsymbol{\theta}^*))}$ is the normalizing constant. Then, $Q(\boldsymbol{\theta})$ is a good approximation of $P(\boldsymbol{\theta})$ in the sense that

$$\ln Q(\boldsymbol{\theta}) + \ln Z_Q = \ln P(\boldsymbol{\theta}^*) + \frac{1}{2}(\boldsymbol{\theta} - \boldsymbol{\theta}^*)^\top (\nabla_{\boldsymbol{\theta}}^2 \ln P(\boldsymbol{\theta}^*))^{-1}(\boldsymbol{\theta} - \boldsymbol{\theta}^*) \cong$$

$$\cong \ln P(\boldsymbol{\theta}^*) + \frac{1}{2}(\boldsymbol{\theta} - \boldsymbol{\theta}^*)^\top (\nabla_{\boldsymbol{\theta}}^2 \ln P(\boldsymbol{\theta}^*))^{-1}(\boldsymbol{\theta} - \boldsymbol{\theta}^*) + \mathcal{O}(\|\boldsymbol{\theta} - \boldsymbol{\theta}^*\|^3) =$$

$$= \ln P(\boldsymbol{\theta}).$$

Notice that $\boldsymbol{\theta}^*$ is in a local maximum, which ensures that the hessian $\nabla_{\boldsymbol{\theta}}^2 \ln P(\boldsymbol{\theta}^*)$ is negative semi-definite. This in turn ensures that the normalizing constant $Z_Q$ exists and that $\text{LAPLACE}_{\max}(\boldsymbol{\theta}^*; P)$ is well defined.

## B.2 If $\boldsymbol{\theta}^*$ is not a local maxima

In order to proceed to a similar derivation when $\nabla_{\boldsymbol{\theta}} \ln P(\boldsymbol{\theta}^*) \neq 0$, we first rearrange the terms in the Taylor expansion. For a more compact notation, we write $\nabla \ln P$ instead of $\nabla_{\boldsymbol{\theta}} \ln P(\boldsymbol{\theta}^*)$ and $\nabla^2 \ln P$ instead of $\nabla_{\boldsymbol{\theta}}^2 \ln P(\boldsymbol{\theta}^*)$.

$$\ln P(\boldsymbol{\theta}) \cong \ln P(\boldsymbol{\theta}^*) + \nabla \ln P(\boldsymbol{\theta} - \boldsymbol{\theta}^*) + \frac{1}{2}(\boldsymbol{\theta} - \boldsymbol{\theta}^*)^\top \nabla^2 \ln P(\boldsymbol{\theta} - \boldsymbol{\theta}^*) =$$

$$= \ln P(\boldsymbol{\theta}^*) - \frac{1}{2}\nabla \ln P^\top \nabla^2 \ln P^{-1} \nabla \ln P$$

$$+ \frac{1}{2}(\boldsymbol{\theta} - \boldsymbol{\theta}^* + \nabla^2 \ln P^{-1} \nabla \ln P)^\top \nabla^2 \ln P(\boldsymbol{\theta} - \boldsymbol{\theta}^* + \nabla^2 \ln P^{-1} \nabla \ln P) =$$

$$= \ln P(\boldsymbol{\theta}^*) - \frac{1}{2}\nabla \ln P^\top \nabla^2 \ln P^{-1} \nabla \ln P + \frac{1}{2}(\boldsymbol{\theta} - \boldsymbol{\theta}_1^*)^\top \nabla^2 \ln P(\boldsymbol{\theta} - \boldsymbol{\theta}_1^*),$$

where we define the new point $\boldsymbol{\theta}_1^*$ as

$$\boldsymbol{\theta}_1^* = \boldsymbol{\theta}^* - (\nabla_{\boldsymbol{\theta}}^2 \ln P(\boldsymbol{\theta}^*))^{-1} \nabla_{\boldsymbol{\theta}} \ln P(\boldsymbol{\theta}^*). \tag{2}$$

Define the Gaussian

$$\text{LAPLACE}(\boldsymbol{\theta}^*; P) := \mathcal{N}\left(\boldsymbol{\theta}|\mu = \boldsymbol{\theta}_1^*, \sigma^2 = -(\nabla_{\boldsymbol{\theta}}^2 \ln P(\boldsymbol{\theta}^*)^{-1}\right), \tag{3}$$

which has density

$$Q(\boldsymbol{\theta}) := \frac{P(\boldsymbol{\theta}^*)}{Z_Q} e^{-\frac{1}{2}\nabla \ln P^\top \nabla^2 \ln P^{-1} \nabla \ln P} e^{\frac{1}{2}(\boldsymbol{\theta}-\boldsymbol{\theta}_1^*)^\top (\nabla_{\boldsymbol{\theta}}^2 \ln P(\boldsymbol{\theta}^*))^{-1}(\boldsymbol{\theta}-\boldsymbol{\theta}_1^*)},$$

where $Z_Q = P(\boldsymbol{\theta}^*)e^{-\frac{1}{2}\nabla \ln P^\top \nabla^2 \ln P^{-1} \nabla \ln P}\sqrt{(-2\pi)^D \det(\nabla_{\boldsymbol{\theta}}^2 \ln P(\boldsymbol{\theta}^*))}$ is the normalizing constant. Then, $Q(\boldsymbol{\theta})$ is a good approximation of $P(\boldsymbol{\theta})$ in the sense that

$$\ln Q(\boldsymbol{\theta}) + \ln Z_Q = \ln P(\boldsymbol{\theta}^*) - \frac{1}{2}\nabla \ln P^\top \nabla^2 \ln P^{-1} \nabla \ln P + \frac{1}{2}(\boldsymbol{\theta}-\boldsymbol{\theta}_1^*)^\top \nabla^2 \ln P(\boldsymbol{\theta}-\boldsymbol{\theta}_1^*) =$$

$$= \ln P(\boldsymbol{\theta}^*) + \nabla \ln P(\boldsymbol{\theta}-\boldsymbol{\theta}^*) + \frac{1}{2}(\boldsymbol{\theta}-\boldsymbol{\theta}^*)^\top \nabla^2 \ln P(\boldsymbol{\theta}-\boldsymbol{\theta}^*) \cong$$

$$\cong \ln P(\boldsymbol{\theta}^*) + \nabla \ln P(\boldsymbol{\theta}-\boldsymbol{\theta}^*) + \frac{1}{2}(\boldsymbol{\theta}-\boldsymbol{\theta}^*)^\top \nabla^2 \ln P(\boldsymbol{\theta}-\boldsymbol{\theta}^*) + \mathcal{O}(\|\boldsymbol{\theta}-\boldsymbol{\theta}^*\|^3) =$$

$$= \ln P(\boldsymbol{\theta}).$$

We highlight four points:

(1) In order to ensure that the normalizing constant $Z_Q$ exists and consequently that LAPLACE($\boldsymbol{\theta}^*; P$) is well defined, the hessian $\nabla_{\boldsymbol{\theta}}^2 \ln P(\boldsymbol{\theta}^*)$ must be negative semi-definite. Differently from the locally maximal case, this is no longer guaranteed.

(2) The method is numerically unstable if $\nabla_{\boldsymbol{\theta}}^2 \ln P(\boldsymbol{\theta}^*)$ has eigenvalues close to 0. This has been empirically observed to commonly be the case (Sagun et al., 2016).

(3) We emphasize that the Taylor expansion is accurate around $\boldsymbol{\theta}^*$, but the Laplace Gaussian is centered in $\boldsymbol{\theta}_1^*$. This is often referred to as "mean shift" and implies that the sampled parameters from the normal distribution over weights are sampled far away from the actual mean.

(4) Now, assume $\ln P(\boldsymbol{\theta})$ to be the composition of a function $f(\boldsymbol{\theta})$ and a loss $l(f)$. If, as in our setting, $f$ is linear and $l$ is concave, then $\ln P(\boldsymbol{\theta})$ is concave, its hessian is guaranteed to be negative semi-definite, and the Laplace approximation is well-defined.

## C  Extended related work

Our proposed method's update rule resembles different methods in the literature. This is rather interesting since, despite arriving at a similar algorithm, they follow different derivations. In this section, we seek to explain the nuances between existing methods and ours.

It is useful to recall our precision update rule

$$\mathbf{H}_{t+1} = (1-\alpha)\mathbf{H}_t + J_{\boldsymbol{\theta}}f_\cdot^t(\boldsymbol{x})^\top \Sigma_{\text{data}}^{-1} J_{\boldsymbol{\theta}}f_\cdot^t(\boldsymbol{x}) \qquad \mathbf{H}_0 = \Sigma_{\text{prior}}^{-1}. \tag{4}$$

To the best of our knowledge, a key conceptual difference with all related works in the literature is **Exact vs. Approximate Hessian:** The second term on the RHS, $J^\top \Sigma^{-1} J$ is identical in all the update formulations but comes from a different derivation. In the formulation of Daxberger et al. (2021) (and others) this is an approximation of the hessian, that *happens to guarantee negative definiteness*. In our formulation, this term comes from the linearization. This term is the exact hessian, i.e. no approximation, such that negative definiteness is implied by the linearization. This results in a more intuitive understanding of the linearization error.

### C.1  Differences with Laplace Redux

Daxberger et al. (2021) specifically develop a post-hoc method, which assumes access to a MAP parameter, nevertheless, they also present an online training scheme. This appears to be grounded in ideas from second-order optimization, while ours is closer linked to the probabilistic model.

Neglecting their prior optimization procedure, Daxberger et al. (2021) considers

$$\mathbf{H}_{t+1} = \Sigma_{\text{prior}}^{-1} + J_{\boldsymbol{\theta}}f_\cdot^t(\boldsymbol{x})^\top \Sigma_{\text{data}}^{-1} J_{\boldsymbol{\theta}}f_\cdot^t(\boldsymbol{x}). \tag{5}$$

We highlight two main differences.

**Iteratively Updating the Hessian vs. Using an Uninformed Prior:** In our formulation the first term of the RHS is the previous precision, which we "discount" with a forgetting term $\alpha$ that comes from the error inherited by moving away from the previous linearization. In the formulation of Daxberger et al. (2021) this is a prior that is assumed to be in the form $\gamma^2 \mathbb{I}$, where the scalar $\gamma$ is optimized at every step through maximization of the evidence (Eq. 6 in their paper). This evidence maximization comes with some drawbacks: (1) It tends to make the Laplace approximation overconfident to outliers. They partially address this issue by adding to the evidence an auxiliary term that depends on an OOD dataset, penalizing it (Eq. 12 in their Appendix). (2) Besides inducing the avoidable need for an OOD dataset, this technique has the same pitfall as VAEs, namely uncertainty should be a derived quantity, not a learned one.

**Computing the hessian at every iteration vs. every epoch:** We update the hessian estimate every iteration. In practice, Daxberger et al. (2021) updates the hessian every epoch. In principle, they could update the hessian more regularly, but to the best of our knowledge, they do not explore this path.

## C.2 Differences with Variational Adaptive Newton Method

Khan et al. (2017) propose the following precision update rule

$$\mathbf{H}_{t+1} = \mathbf{H}_t + \alpha J_{\boldsymbol{\theta}} f_{\cdot}^t(\boldsymbol{x})^\top \Sigma_{\text{data}}^{-1} J_{\boldsymbol{\theta}} f_{\cdot}^t(\boldsymbol{x}). \tag{6}$$

The updates are very similar to ours, where the only difference is that the scaling is made on the Jacobian product instead of on the previous precision $\mathbf{H}_t$. Both their and our updates are "additive" in the sense that the magnitude is increasing, for them strictly monotonically, for us on average depending on the magnitude of $\alpha$. Thus, in both cases, the variance will approach 0 in the limit, modeling epistemic uncertainty disappearing for infinitely long training.

Their update rule comes from the Variational Optimization setting, which can be viewed as an instance of Variational Inference neglecting the KL term. They highlight strong similarities with Newton's method.

## C.3 Differences with Noisy Natural Gradient

Zhang et al. (2017) propose the precision update rule

$$\mathbf{H}_{t+1} = (1 - \alpha)\mathbf{H}_t + \alpha J_{\boldsymbol{\theta}} f_{\cdot}^t(\boldsymbol{x})^\top \Sigma_{\text{data}}^{-1} J_{\boldsymbol{\theta}} f_{\cdot}^t(\boldsymbol{x}). \tag{7}$$

Again, the update rule is very similar to ours, besides the scaling being applied to the Jacobian product. They use a convex sum, which makes the update rule "norm preserving". Thus, these updates have very different asymptotic behavior than ours.

Their update rule comes from applying, in the context of Variational Inference, natural gradient to the variational posterior distribution, instead of directly on the parameters space. The natural gradient is deeply connected with LAE and natural parameters highlight the importance of updating the precision matrix instead of the covariance.

They also extend their derivation to the KFAC approximation of the hessian (in place of the exact diagonal), this is made through the use of matrix variate Gaussian. Despite not being considered in this work, a similar derivation is in principle feasible in our setting.

## C.4 Connection with Adam

Both Zhang et al. (2017) and Khan et al. (2017) highlight strong similarities with the Adam method (Kingma and Ba, 2015a). We share these similarities and we highlight them too. The "connection point" is a noisy version of Adam. Zhang et al. (2017) describe this method and call it "Noisy Adam" (Algorithm 1 in their paper). The difference from the vanilla version is that at each step, instead of using the current parameter, they use a noisy version of it. The noise magnitude is the pseudo-second-order term ($v_t$ in the original Adam paper (Kingma and Ba, 2015b)).

We can then interpret Noisy Adam as an instance of our variational setting, specifically where the expectation estimate $\mathbb{E}_{q(\theta)}[\cdot]$ is made through Monte Carlo estimation with $N = 1$ samples. Having the methods in the same setting, we can compare them and highlight the two main differences.

First, we emphasize the difference is in the second order term. Similarly to Zhang et al. (2017) and Khan et al. (2017), we use the diagonal of the hessian (technically the diagonal of the GGN), while Adam uses the pointwise square of the gradient, which they call the second raw moment and is intended as a cheaply computable approximation of the Hessian.

Another difference is that Adam applies the square root. This is a minor point since, as pointed out by Zhang et al. (2017), this change may affect optimization performance, but does not change the fixed points.

### C.5 Connection with Bayes by Backpropagation

Bayes by Backprop (Blundell et al., 2015) can be viewed as a sample-based approximation of Laplace. In order to show this relation, we recall two very powerful equations (Opper and Archambeau, 2009). Specifically, let $\mu$, $\Sigma$ be the parameter of a Gaussian distribution $q(\theta) \sim \mathcal{N}(\mu, \Sigma)$, and let $V(\theta)$ be an arbitrary $L^2$ integrable function (the log-likelihood in our case). Then, we are interested in the derivatives of $\mathbb{E}_{\theta \sim q}[V(\theta)]$. By standard Fourier analysis and integration by part, we have

$$\nabla_\mu \mathbb{E}_{\theta \sim q}[V(\theta)] = \mathbb{E}_{\theta \sim q}[\nabla_\theta V(\theta)], \tag{8}$$

$$\nabla_\Sigma \mathbb{E}_{\theta \sim q}[V(\theta)] = \frac{1}{2}\mathbb{E}_{\theta \sim q}[\nabla_\theta^2 V(\theta)]. \tag{9}$$

While the first equation is somehow trivial, the second highlight a very deep relationship. The LHS can be rewritten as

$$\nabla_\Sigma \mathbb{E}_{\theta \sim q}[V(\theta)] = \nabla_\Sigma \mathbb{E}_{\epsilon \sim \mathcal{N}(0,1)}[V(\mu + \epsilon\Sigma)] = \mathbb{E}_{\epsilon \sim \mathcal{N}(0,1)}[\nabla_\Sigma V(\mu + \epsilon\Sigma)]. \tag{10}$$

We can recognize that inside the expectation on the RHS is exactly the update rule for the variance in the bayes by backprop method (Blundell et al., 2015). Thus, we can interpret bayes by backprop as a one-sample Monte Carlo estimation of the expected value. The equations shows that in the limit of infinite samples, the bayes by backprop update step $\nabla_\Sigma V(\mu + \epsilon\Sigma)$ converges to the (averaged) Laplace approximation step $\mathbb{E}_{\theta \sim q}[\nabla_\theta^2 V(\theta)]$ up to a factor 2. This motivates both the power and the instability of bayes by backprop.

## D  Model

### D.1  Overview

Let $\mathbb{X} = \mathbb{R}^D$ be the *data* space, let $\mathbb{Y} = \mathbb{R}^D$ be the *reconstruction* space and let $\Theta = \mathbb{R}^P$ be the *parameter* space. Let $\mathbb{F} = (\Theta \to (\mathbb{X} \to \mathbb{Y}))$ be the space of operators from $\Theta$ to the space of operators from $\mathbb{X}$ to $\mathbb{Y}$. We will denote a function $f \in \mathbb{F}$ applied to a parameter $\boldsymbol{\theta} \in \Theta$ as $f_{\boldsymbol{\theta}} : \mathbb{X} \mapsto \mathbb{Y}$. This will represent, for example, a NN $f$ with a specific set of parameter (weights) $\boldsymbol{\theta} \in \Theta$, that maps some data $\boldsymbol{x} \in \mathbb{X}$ to some reconstruction $f_{\boldsymbol{\theta}}(\boldsymbol{x}) = \boldsymbol{y} \in \mathbb{Y}$.

Let $\mathbb{X} \times \mathbb{Y} \times \Theta \times \mathbb{F}$ be a probability space. The only assumption we make on this space is

$$p(\boldsymbol{y}|\boldsymbol{x}, \boldsymbol{\theta}, f) \sim \mathcal{N}(\boldsymbol{y}|\mu = f_{\boldsymbol{\theta}}(\boldsymbol{x}), \sigma^2 = \Sigma) \qquad \forall (\boldsymbol{x}, \boldsymbol{\theta}, f) \in \mathbb{X} \times \Theta \times \mathbb{F} \tag{11}$$

where $\Sigma \in \mathfrak{M}(\mathbb{R}^D \times \mathbb{R}^D)$ is a *fixed* variance matrix. This is a common assumption for regression tasks, and is sometimes referred to as the "data noise" or "reconstruction error". In this paper, we fix $\Sigma = \mathbb{I}$, but the derivations hold for the general case. With only this assumption, the distribution is undefined and multiple solutions can exist. Thus, we require more assumptions.

A dataset $\mathcal{D} = \{\boldsymbol{x}_n\}$ is a finite of infinite collection of data $\boldsymbol{x}_n \in \mathbb{X}$ that is assumed to follow a certain, fixed but unknown, distribution

$$\boldsymbol{x}_n \sim p(\boldsymbol{x}). \tag{12}$$

Sacrificing slim notation for the sake of clarity, we introduce an operator $\mathcal{I} : \mathbb{X} \mapsto \mathbb{Y}$. This represents the ideal reconstruction for a given $\boldsymbol{x}$. In the standard supervised setting, this would be the operator (defined on the dataset only) that maps each data input to its label. In our unsupervised setting, where $\mathbb{X}$ and $\mathbb{Y}$ are *the same space*, the operator $\mathcal{I}$ is simply the identity (Indeed they are *not* the same space, they are isomorphic spaces that we identify through the operator $\mathcal{I}$). Since $\mathcal{I}$ is the identity, it is often

neglected in the literature, which can lead to unclear and potentially ambiguous Bayesian derivations. Thus, we choose to adopt this heavier, but more precise notation.

We assume access to a specific $f^{NN} \in \mathbb{F}$. Practically this will be our NN architecture, i.e. an operator that, given a set of parameters $\boldsymbol{\theta} \in \Theta$ gives rise to a function from $\mathbb{X}$ to $\mathbb{Y}$. Having $f^{NN}$ fixed, one may consider $f$ not to be stochastic anymore, we choose to still explicitly condition on $f$ in order to have a clearer notation in later stages. Note that, despite not being covered in this work, a proper stochastic derivation also on the NN architecture should be feasible.

## D.2 Objective

The NN's parameter optimization process in this full Bayesian probabilistic framework can be viewed as: given a fixed $f^{NN} \in \mathbb{F}$, namely the NN architecture, maximise the reconstruction probability of $\mathcal{I}(\boldsymbol{x}_n)$ over the dataset $\mathcal{D}$

$$\mathbb{E}_{\boldsymbol{x}_n \sim p(\boldsymbol{x})}\left[p(\boldsymbol{y}|\boldsymbol{x}_n, f^{NN})\big|_{\boldsymbol{y}=\mathcal{I}(\boldsymbol{x}_n)}\right] = \sum_{\boldsymbol{x}_n \in \mathcal{D}} p(\boldsymbol{y}|\boldsymbol{x}_n, f^{NN})\big|_{\boldsymbol{y}=\mathcal{I}(\boldsymbol{x}_n)}, \tag{13}$$

where the untractable $p(\boldsymbol{y}|\boldsymbol{x}_n, f^{NN})$ can be expanded in $\boldsymbol{\theta}$ and thus related to our hypothesis (11) as

$$p(\boldsymbol{y}|\boldsymbol{x}_n, f^{NN}) = \mathbb{E}_{\boldsymbol{\theta} \sim p(\boldsymbol{\theta}|\boldsymbol{x}_n, f^{NN})}\left[p(\boldsymbol{y}|\boldsymbol{x}_n, \boldsymbol{\theta}, f^{NN})\right]. \tag{14}$$

Notice that the only unfixed quantity is the distribution on the parameters, which we will optimize for. We are not interested in finding a datapoint-dependant distribution, but rather one that maximise all reconstructions at the same time, i.e. $p(\boldsymbol{\theta}|f^{NN}) = p(\boldsymbol{\theta}|\boldsymbol{x}_n, f^{NN})$. We can then frame Bayesian optimization as: find a distribution on parameters such that

$$q(\boldsymbol{\theta}) \in \arg \max_{p(\boldsymbol{\theta}|f^{NN})} \sum_{\boldsymbol{x}_n \in \mathcal{D}} \mathbb{E}_{\boldsymbol{\theta} \sim p(\boldsymbol{\theta}|f^{NN})}\left[p(\boldsymbol{y}|\boldsymbol{x}_n, \boldsymbol{\theta}, f^{NN})\big|_{\boldsymbol{y}=\mathcal{I}(\boldsymbol{x}_n)}\right] \tag{15}$$

$$= \arg \max_{p(\boldsymbol{\theta}|f^{NN})} \sum_{\boldsymbol{x}_n \in \mathcal{D}} \mathbb{E}_{\boldsymbol{\theta} \sim p(\boldsymbol{\theta}|f^{NN})}\left[p(\mathcal{I}(\boldsymbol{x}_n)|\mathcal{N}(f_{\boldsymbol{\theta}}^{NN}(\boldsymbol{x}_n), \Sigma))\right]. \tag{16}$$

Moreover, finding this optimum in the space $\Delta(\Theta)$ of all distributions on $\Theta$ is not tractable. So, as commonly done, we restrict ourselves to the subset $\mathcal{G}(\Theta) \subset \Delta(\Theta)$ of Gaussians over $\Theta$. Then, a *solution* in our context is

$$q(\boldsymbol{\theta}) \in \arg \max_{q \in \mathcal{G}(\Theta)} \sum_{\boldsymbol{x}_n \in \mathcal{D}} \mathbb{E}_{\boldsymbol{\theta} \sim q(\boldsymbol{\theta})}\left[p(\mathcal{I}(\boldsymbol{x}_n)|\mathcal{N}(f_{\boldsymbol{\theta}}^{NN}(\boldsymbol{x}_n), \Sigma))\right]. \tag{17}$$

We emphasize that this solution has no guarantees of being unique, but we are interested in finding one of them.

## D.3 Joint distribution for a fixed datapoint

Let us first get a better understanding of the joint distribution on $\mathbb{Y} \times \Theta$ conditional to a fixed datapoint $\boldsymbol{x} \in \mathbb{X}$ and a network architecture $f \in \mathbb{F}$

$$p(\boldsymbol{y}, \boldsymbol{\theta}|\boldsymbol{x}, f). \tag{18}$$

This distribution has two *marginals*

$$p(\boldsymbol{y}|\boldsymbol{x}, f) \tag{19}$$

$$p(\boldsymbol{\theta}|\boldsymbol{x}, f) \tag{20}$$

and two *conditionals*

$$p(\boldsymbol{y}|\boldsymbol{\theta}, \boldsymbol{x}, f) \tag{21}$$

$$p(\boldsymbol{\theta}|\boldsymbol{y}, \boldsymbol{x}, f). \tag{22}$$

These four quantities must satisfy the system of two "recursive" equations

$$p(\boldsymbol{y}|\boldsymbol{x}, f) = \int_{\Theta} p(\boldsymbol{y}|\boldsymbol{\theta}, \boldsymbol{x}, f)p(\boldsymbol{\theta}|\boldsymbol{x}, f)\mathrm{d}\boldsymbol{\theta} = \tag{23}$$

$$= \mathbb{E}_{\boldsymbol{\theta} \sim p(\boldsymbol{\theta}|\boldsymbol{x}, f)}[p(\boldsymbol{y}|\boldsymbol{\theta}, \boldsymbol{x}, f)]$$

$$p(\boldsymbol{\theta}|\boldsymbol{x}, f) = \int_{\mathbb{Y}} p(\boldsymbol{\theta}|\boldsymbol{y}, \boldsymbol{x}, f)p(\boldsymbol{y}|\boldsymbol{x}, f)\mathrm{d}\boldsymbol{y} = \tag{24}$$

$$= \mathbb{E}_{\boldsymbol{y} \sim p(\boldsymbol{y}|\boldsymbol{x}, f)}[p(\boldsymbol{\theta}|\boldsymbol{y}, \boldsymbol{x}, f)].$$

If these are satisfied then the joint is a well-defined distribution and we can apply Bayes rule

$$p(\boldsymbol{y}|\boldsymbol{\theta}, \boldsymbol{x}, f)p(\boldsymbol{\theta}|\boldsymbol{x}, f) = p(\boldsymbol{y}, \boldsymbol{\theta}|\boldsymbol{x}, f) = p(\boldsymbol{\theta}|\boldsymbol{y}, \boldsymbol{x}, f)p(\boldsymbol{y}|\boldsymbol{x}, f) \tag{25}$$

which in logarithmic form is

$$\log p(\boldsymbol{y}|\boldsymbol{\theta}, \boldsymbol{x}, f) + \log p(\boldsymbol{\theta}|\boldsymbol{x}, f) = \log p(\boldsymbol{\theta}|\boldsymbol{y}, \boldsymbol{x}, f) + \log p(\boldsymbol{y}|\boldsymbol{x}, f). \tag{26}$$

We can factor in the assumptions. The "data noise" *assumption* gives us one of the two conditionals:

$$p(\boldsymbol{y}|\boldsymbol{\theta}, \boldsymbol{x}, f) \sim \mathcal{N}(\boldsymbol{y}|\mu = f_{\boldsymbol{\theta}}(\boldsymbol{x}), \sigma^2 = \Sigma). \tag{27}$$

The "Gaussian parameter" *assumption* gives us one of the two marginals:

$$p(\boldsymbol{\theta}|\boldsymbol{x}, f) = q^t(\boldsymbol{\theta}) \sim \mathcal{N}(\boldsymbol{\theta}|\mu = \boldsymbol{\theta}_t, \sigma^2 = \mathbf{H}_t^{-1}). \tag{28}$$

With these in place, the joint distribution is uniquely defined. The other marginal, by Eq. 23, is

$$p(\boldsymbol{y}|\boldsymbol{x}, f) = \mathbb{E}_{\boldsymbol{\theta} \sim q^t(\boldsymbol{\theta})}[p(\boldsymbol{y}|\mathcal{N}(f_{\boldsymbol{\theta}}(\boldsymbol{x}), \Sigma))] \tag{29}$$

and the other conditional, by Bayes rule, is

$$p(\boldsymbol{\theta}|\boldsymbol{y}, \boldsymbol{x}, f) = \frac{p(\boldsymbol{y}|\boldsymbol{\theta}, \boldsymbol{x}, f)p(\boldsymbol{\theta}|\boldsymbol{x}, f)}{p(\boldsymbol{y}|\boldsymbol{x}, f)}. \tag{30}$$

Despite being uniquely defined, the integral in Eq. 29 is, with a general $f$, intractable, and so is the joint distribution.

**But why do we even care?** The intractability of $p(\boldsymbol{y}|\boldsymbol{x}, f)$ in Eq. 29 may at first glance appear irrelevant. This is the case, for instance, with bayes by backprop (Blundell et al., 2015) methods. They simply need access to the gradient of this quantity. For this purpose, a simple Monte Carlo estimate of the expectation is enough.

On the other hand, we are interested in recovering a meaningful distribution on parameters. This imply that we aim at using Eq. 29 to enforce that Eq. 24 holds. For this purpose we need access to the the density $p(\boldsymbol{y}|\boldsymbol{x}, f)$, so a Monte Carlo estimate of Eq. 29 is not enough.

### D.4   Linear $f$

**Theorem 1.** *Given the data noise assumption from Eq. 27*

$$p(\boldsymbol{y}|\boldsymbol{\theta}, \boldsymbol{x}, f) \sim \mathcal{N}(\boldsymbol{y}|\mu = f_{\boldsymbol{\theta}}(\boldsymbol{x}), \sigma^2 = \Sigma), \tag{31}$$

*given the Gaussian parameter assumption from Eq. 28 for some $\boldsymbol{\theta}_t$, $\mathbf{H}_t$*

$$p(\boldsymbol{\theta}|\boldsymbol{x}, f) = q^t(\boldsymbol{\theta}) \sim \mathcal{N}(\boldsymbol{\theta}|\mu = \boldsymbol{\theta}_t, \sigma^2 = \mathbf{H}_t^{-1}), \tag{32}$$

*assume that $f$ is linear in $\boldsymbol{\theta}$, i.e.*

$$f_{\boldsymbol{\theta}}(\boldsymbol{x}) = f_0(\boldsymbol{x}) + Jf.(\boldsymbol{x})\boldsymbol{\theta} \qquad \forall \boldsymbol{\theta} \in \Theta, \forall \boldsymbol{x} \in \mathbb{X}. \tag{33}$$

*Then the joint distribution is Gaussian itself*

$$p(\boldsymbol{\theta}, \boldsymbol{y}|\boldsymbol{x}, f) \sim \mathcal{N}((\boldsymbol{\theta}, \boldsymbol{y})|\mu_t, \Sigma_t) \tag{34}$$

*where*

$$\mu_t = \begin{pmatrix} \boldsymbol{\theta}_t \\ f_{\boldsymbol{\theta}_t}(\boldsymbol{x}) \end{pmatrix}, \text{ and } \Sigma_t = \begin{pmatrix} \mathbf{H}_t^{-1} & \Sigma Jf.(\boldsymbol{x})^\top \\ Jf.(\boldsymbol{x})\Sigma & \left(Jf.(\boldsymbol{x})^\top \mathbf{H}_t Jf.(\boldsymbol{x})\right)^{-1} + \Sigma \end{pmatrix}.$$

*Proof.* With the further assumption of linearity of $f$, we can explicitly carry out the integral in the expectation in Eq. 29

$$p(\boldsymbol{y}|\boldsymbol{x}, f) = \mathbb{E}_{\boldsymbol{\theta} \sim q^t(\boldsymbol{\theta})}[p(\boldsymbol{y}|\mathcal{N}(f_{\boldsymbol{\theta}}(\boldsymbol{x}), \Sigma))] =$$

$$= \int_{\Theta} p(\boldsymbol{y}|\mathcal{N}(f_{\boldsymbol{\theta}}(\boldsymbol{x}), \Sigma))q^t(\boldsymbol{\theta})\mathrm{d}\boldsymbol{\theta} =$$

$$= \int_{\Theta} p(\boldsymbol{y}|\mathcal{N}(f_{\boldsymbol{\theta}}(\boldsymbol{x}), \Sigma))p(\boldsymbol{\theta}|\mathcal{N}(\boldsymbol{\theta}_t, \mathbf{H}_t^{-1}))\mathrm{d}\boldsymbol{\theta} =$$

$$= \int_{\Theta} p(\boldsymbol{y}|\mathcal{N}(f_0(\boldsymbol{x}) + Jf.(\boldsymbol{x})\boldsymbol{\theta}, \Sigma))p(\boldsymbol{\theta}|\mathcal{N}(\boldsymbol{\theta}_t, \mathbf{H}_t^{-1}))\mathrm{d}\boldsymbol{\theta} =$$

$$= p(\boldsymbol{y}|\mathcal{N}(f_{\boldsymbol{\theta}_t}(\boldsymbol{x}), (Jf.(\boldsymbol{x})^\top \mathbf{H}_t Jf.(\boldsymbol{x}))^{-1} + \Sigma)).$$

We emphasize that as a consequence $\nabla_{\boldsymbol{\theta}}^2 \log p(\boldsymbol{y}|\boldsymbol{x}, f)$ is not dependent on $\boldsymbol{\theta}_t$. $\qquad\square$

Having Theorem 1 in place, we can go back to our original problem. We need to deal with a fixed *non-linear* architecture $f^{NN}$. We can exploit Theorem 1 by defining $f^t$: a linearization of $f^{NN}$ with a first-order Taylor expansion around $\boldsymbol{\theta}_t$

$$f^t_{\boldsymbol{\theta}}(\boldsymbol{x}) := \text{TAYLOR}(f^{NN}, \boldsymbol{\theta}_t)(\boldsymbol{x}) =$$
$$= f^{NN}_{\boldsymbol{\theta}_t}(\boldsymbol{x}) + J_{\boldsymbol{\theta}} f^{NN}_{\boldsymbol{\theta}_t}(\boldsymbol{x})(\boldsymbol{\theta} - \boldsymbol{\theta}_t) \tag{35}$$

and it holds that

$$f^{NN}_{\boldsymbol{\theta}}(\boldsymbol{x}) = f^t_{\boldsymbol{\theta}}(\boldsymbol{x}) + \mathcal{O}(\|\boldsymbol{\theta} - \boldsymbol{\theta}_t\|^2). \tag{36}$$

Recalling Eq. 11 both for $f^{NN}$ and for $f^t$

$$p(\boldsymbol{y}|\boldsymbol{x}, \boldsymbol{\theta}, f^{NN}) \sim \mathcal{N}(\boldsymbol{y}|\mu = f^{NN}_{\boldsymbol{\theta}}(\boldsymbol{x}), \sigma^2 = \Sigma) \tag{37}$$

$$p(\boldsymbol{y}|\boldsymbol{x}, \boldsymbol{\theta}, f^t) \sim \mathcal{N}(\boldsymbol{y}|\mu = f^t_{\boldsymbol{\theta}}(\boldsymbol{x}), \sigma^2 = \Sigma) \tag{38}$$

that, together with Eq. 36, imply

$$\mathcal{N}(\boldsymbol{y}|\mu = f^{NN}_{\boldsymbol{\theta}}(\boldsymbol{x}), \sigma^2 = \Sigma) \sim \mathcal{N}(\boldsymbol{y}|\mu = f^t_{\boldsymbol{\theta}}(\boldsymbol{x}) + \mathcal{O}(\|\boldsymbol{\theta} - \boldsymbol{\theta}_t\|^2), \sigma^2 = \Sigma), \tag{39}$$

where we can interpret the unknown $\mathcal{O}(\|\boldsymbol{\theta} - \boldsymbol{\theta}_t\|^2)$ as $\boldsymbol{\theta}$-dependent noise. More specifically, calling $\gamma > 0$ the scalar constant of the $\mathcal{O}$-term, we assume that

$$\mathcal{O}(\|\boldsymbol{\theta} - \boldsymbol{\theta}_t\|^2) \approx \epsilon(\boldsymbol{\theta} - \boldsymbol{\theta}_t)^2 \qquad \text{where } \epsilon \sim \mathcal{N}(0, \gamma\mathbb{I}) \tag{40}$$

and thus, from Eq. 39, we have

$$\mathcal{N}(\boldsymbol{y}|\mu = f^{NN}_{\boldsymbol{\theta}}(\boldsymbol{x}), \sigma^2 = \Sigma) \sim \mathcal{N}(\boldsymbol{y}|\mu = f^t_{\boldsymbol{\theta}}(\boldsymbol{x}), \sigma^2 = \Sigma + \gamma\|\boldsymbol{\theta} - \boldsymbol{\theta}_t\|^2\mathbb{I}). \tag{41}$$

At this point, integrals are not analytically tractable, and thus a proper proof is not feasible, the intuition is that this increased variance reflects in increased variance in $p(\boldsymbol{\theta}|\boldsymbol{x}, f^{NN})$

$$\nabla^2_{\boldsymbol{\theta}} \log p(\boldsymbol{\theta}|\boldsymbol{x}, f^{NN})\big|_{\boldsymbol{\theta}=\boldsymbol{\theta}_{t+1}} \approx \nabla^2_{\boldsymbol{\theta}} \log p(\boldsymbol{\theta}|\boldsymbol{x}, f^t)\big|_{\boldsymbol{\theta}=\boldsymbol{\theta}_{t+1}} + \gamma\|\boldsymbol{\theta}_{t+1} - \boldsymbol{\theta}_t\|^2, \tag{42}$$

where we introduce a hyperparameter $\alpha > 0$ to cope with this added variance. If we then assume the Jacobian $J_{\boldsymbol{\theta}} f^{NN}$ to be a Lipschitz function, then the Lipschitz constant is an upper bound on $\gamma$, as follows from the Taylor expansion of Eq. (35). If this Lipschitz constant is smaller than the inverse of the gradient step $1/\|\boldsymbol{\theta}_{t+1} - \boldsymbol{\theta}_t\|$ (which is not an unreasonable assumption for the gradient ascent to be stable) we have

$$\gamma\|\boldsymbol{\theta}_{t+1} - \boldsymbol{\theta}_t\|^2 \approx \|\boldsymbol{\theta}_{t+1} - \boldsymbol{\theta}_t\| \tag{43}$$

that gives us a plausible order of magnitude for choosing the hyperparameter $\alpha$.

**Motivation:** During training, we produce a sequence of Gaussians $q^t(\boldsymbol{\theta}) \sim \mathcal{N}(\boldsymbol{\theta}_t, \mathbf{H}_t^{-1})$ that we assume to be the distribution $p(\boldsymbol{\theta}|\boldsymbol{x}, f^t)$, at every step $t \geq 0$. This distribution $q^t$ is then used for (1) Gaussian derivation in the linear case, for (2) Monte Carlo sampling in the update rule $\boldsymbol{\theta}_t \to \boldsymbol{\theta}_{t+1}$ and (3), as second order derivative, for update rule $\mathbf{H}_t \to \mathbf{H}_{t+1}$.

Moreover, given that this distribution $q^t$ is our "best guess so far", we assume it to be also the distribution $p(\boldsymbol{\theta}|\boldsymbol{x}, f^{NN})$. This, being $f^{NN}$ not linear, (1) cannot be used for Gaussian derivation, (2) can reasonably be used for sampling (and thus we derive the improved update rule Eq. (52)), and (3) can somehow be used as second order derivative (and thus we derive the improved update rule Eq. (53)), but the latter requires some more care. That is why we introduce the parameter $\alpha$.

### D.5 Iterative learning

Our learning method produces a sequence of Gaussians

$$q^t(\boldsymbol{\theta}) \sim \mathcal{N}(\boldsymbol{\theta}|\mu = \boldsymbol{\theta}_t, \sigma^2 = \mathbf{H}_t^{-1}) \tag{44}$$

and a sequence of linearized functions

$$f^t = \text{TAYLOR}(f^{NN}, \boldsymbol{\theta}_t) \tag{45}$$

for every $t \geq 0$.

**Initialization** is trivially done using a Gaussian prior on the parameters.

$$\boldsymbol{\theta}_0 = \boldsymbol{\theta}^{\text{prior}} \qquad \mathbf{H}_0 = (\Sigma^{\text{prior}})^{-1} \tag{46}$$

**Iterative step** is made in two steps. First, having access to $\boldsymbol{\theta}_t$, we "generate" the linearization $f^t$. Practically this is equivalent to computing the two quantities $f_{\boldsymbol{\theta}_t}^{NN}(\boldsymbol{x})$ and $J_{\boldsymbol{\theta}} f_{\boldsymbol{\theta}_t}^{NN}(\boldsymbol{x})$, that, together with the value $\boldsymbol{\theta}_t$ are actually equivalent to "generating" $f^t$, as Eq. 35 shows.

Second, we compute the Gaussian parameters $\boldsymbol{\theta}_{t+1}$ and $\mathbf{H}_{t+1}$.

Recalling our aim of maximizing the quantity Eq. (17), update on $q(\cdot)$ means, $\boldsymbol{\theta}_t \to \boldsymbol{\theta}_{t+1}$, is *ideally* made through gradient ascent steps on $p(\boldsymbol{y}|\boldsymbol{x}, f)|_{\boldsymbol{y}=\mathcal{I}(\boldsymbol{x})}$. As this is intractable, we instead do gradient steps on the lower bound $\mathcal{L}_{\boldsymbol{y}}$ of (the log of) Eq. 23

$$\boldsymbol{\theta}_{t+1} = \boldsymbol{\theta}_t + \lambda \nabla_{\boldsymbol{\theta}} \mathcal{L}_{\boldsymbol{y}}\Big|_{\boldsymbol{y}=\mathcal{I}(\boldsymbol{x})} \tag{47}$$

where

$$\mathcal{L}_{\boldsymbol{y}} = \mathbb{E}_{\boldsymbol{\theta} \sim p(\boldsymbol{\theta}|\boldsymbol{x}, f^t)}[\log p(\boldsymbol{y}|\boldsymbol{\theta}, \boldsymbol{x}, f^t)] \leq \log \mathbb{E}_{\boldsymbol{\theta} \sim p(\boldsymbol{\theta}|\boldsymbol{x}, f^t)}[p(\boldsymbol{y}|\boldsymbol{\theta}, \boldsymbol{x}, f^t)] \tag{48}$$

and so

$$\nabla_{\boldsymbol{\theta}} \mathcal{L}_{\boldsymbol{y}}\Big|_{\boldsymbol{y}=\mathcal{I}(\boldsymbol{x})} = \mathbb{E}_{\boldsymbol{\theta} \sim p(\boldsymbol{\theta}|\boldsymbol{x}, f^t)} \left[\nabla_{\boldsymbol{\theta}} \log p(\boldsymbol{y}|\boldsymbol{\theta}, \boldsymbol{x}, f^t)\Big|_{\boldsymbol{y}=\mathcal{I}(\boldsymbol{x})}\right] =$$
$$= \mathbb{E}_{\boldsymbol{\theta} \sim q^t(\boldsymbol{\theta})} \left[\nabla_{\boldsymbol{\theta}} \log p(\mathcal{I}(\boldsymbol{x})|\mathcal{N}(f_{\boldsymbol{\theta}}^t(\boldsymbol{x}), \Sigma))\right] . \tag{49}$$

Recalling the Laplace approximation Eq. (3), the negative precision, $-\mathbf{H}_{t+1}$, is *ideally* set to be the the hessian of the log probability $p(\boldsymbol{\theta}|\boldsymbol{x}, f)$, evaluated in $\boldsymbol{\theta}_{t+1}$. As this is intractable we instead set it to the hessian of the lower bound $\mathcal{L}_{\boldsymbol{\theta}}$ of (the log of) Eq. (24)

$$\mathbf{H}_{t+1} = -\nabla_{\boldsymbol{\theta}}^2 \mathcal{L}_{\boldsymbol{\theta}}\Big|_{\boldsymbol{\theta}=\boldsymbol{\theta}_{t+1}} \tag{50}$$

where

$$\mathcal{L}_{\boldsymbol{\theta}} = \mathbb{E}_{\boldsymbol{y} \sim p(\boldsymbol{y}|\boldsymbol{x}, f^t)}[\log p(\boldsymbol{\theta}|\boldsymbol{y}, \boldsymbol{x}, f^t)] \leq \log \mathbb{E}_{\boldsymbol{y} \sim p(\boldsymbol{y}|\boldsymbol{x}, f^t)}[p(\boldsymbol{\theta}|\boldsymbol{y}, \boldsymbol{x}, f^t)] \tag{51}$$

and so

$$\nabla_{\boldsymbol{\theta}}^2 \mathcal{L}_{\boldsymbol{\theta}}\Big|_{\boldsymbol{\theta}=\boldsymbol{\theta}_{t+1}} = \mathbb{E}_{\boldsymbol{y} \sim p(\boldsymbol{y}|\boldsymbol{x}, f^t)} \left[\nabla_{\boldsymbol{\theta}}^2 \log p(\boldsymbol{\theta}|\boldsymbol{y}, \boldsymbol{x}, f^t)\Big|_{\boldsymbol{\theta}=\boldsymbol{\theta}_{t+1}}\right]$$

via Eq. (26)

$$= \mathbb{E}_{\boldsymbol{y} \sim p(\boldsymbol{y}|\boldsymbol{x}, f^t)} \left[\nabla_{\boldsymbol{\theta}}^2 \log p(\boldsymbol{y}|\boldsymbol{\theta}, \boldsymbol{x}, f^t)\Big|_{\boldsymbol{\theta}=\boldsymbol{\theta}_{t+1}} + \nabla_{\boldsymbol{\theta}}^2 \log p(\boldsymbol{\theta}|\boldsymbol{x}, f^t)\Big|_{\boldsymbol{\theta}=\boldsymbol{\theta}_{t+1}} + \right.$$
$$\left. - \nabla_{\boldsymbol{\theta}}^2 \log p(\boldsymbol{y}|\boldsymbol{x}, f^t)\Big|_{\boldsymbol{\theta}=\boldsymbol{\theta}_{t+1}}\right]$$

via Theorem 1

$$= \mathbb{E}_{\boldsymbol{y} \sim p(\boldsymbol{y}|\boldsymbol{x}, f^t)} \left[\nabla_{\boldsymbol{\theta}}^2 \log p(\boldsymbol{y}|\boldsymbol{\theta}, \boldsymbol{x}, f^t)\Big|_{\boldsymbol{\theta}=\boldsymbol{\theta}_{t+1}} + \nabla_{\boldsymbol{\theta}}^2 \log p(\boldsymbol{\theta}|\boldsymbol{x}, f^t)\Big|_{\boldsymbol{\theta}=\boldsymbol{\theta}_{t+1}}\right]$$

via the chain rule Eq. (55)

$$= \mathbb{E}_{\boldsymbol{y} \sim p(\boldsymbol{y}|\boldsymbol{x}, f^t)} \left[J_{\boldsymbol{\theta}} f_{.}^t(\boldsymbol{x})^\top \nabla_{\boldsymbol{y}}^2 \log p(\boldsymbol{y}|\boldsymbol{\theta}_{t+1}, \boldsymbol{x}, f^t) J_{\boldsymbol{\theta}} f_{.}^t(\boldsymbol{x}) + \nabla_{\boldsymbol{\theta}}^2 \log p(\boldsymbol{\theta}|\boldsymbol{x}, f^t)\Big|_{\boldsymbol{\theta}=\boldsymbol{\theta}_{t+1}}\right]$$

via HP Eq. (11)

$$= \mathbb{E}_{\boldsymbol{y} \sim p(\boldsymbol{y}|\boldsymbol{x}, f^t)} \left[-J_{\boldsymbol{\theta}} f_{.}^t(\boldsymbol{x})^\top \Sigma^{-1} J_{\boldsymbol{\theta}} f_{.}^t(\boldsymbol{x}) + \nabla_{\boldsymbol{\theta}}^2 \log p(\boldsymbol{\theta}|\boldsymbol{x}, f^t)\Big|_{\boldsymbol{\theta}=\boldsymbol{\theta}_{t+1}}\right]$$

$$= -J_{\boldsymbol{\theta}} f_{.}^t(\boldsymbol{x})^\top \Sigma^{-1} J_{\boldsymbol{\theta}} f_{.}^t(\boldsymbol{x}) + \nabla_{\boldsymbol{\theta}}^2 \log p(\boldsymbol{\theta}|\boldsymbol{x}, f^t)\Big|_{\boldsymbol{\theta}=\boldsymbol{\theta}_{t+1}}$$

$$= -J_{\boldsymbol{\theta}} f_{.}^t(\boldsymbol{x})^\top \Sigma^{-1} J_{\boldsymbol{\theta}} f_{.}^t(\boldsymbol{x}) + \nabla_{\boldsymbol{\theta}}^2 \log q^t(\boldsymbol{\theta})\Big|_{\boldsymbol{\theta}=\boldsymbol{\theta}_{t+1}}$$

$$= -J_{\boldsymbol{\theta}} f_{.}^t(\boldsymbol{x})^\top \Sigma^{-1} J_{\boldsymbol{\theta}} f_{.}^t(\boldsymbol{x}) - \mathbf{H}_t .$$

### D.5.1 Improved update rule

As said, the update $\boldsymbol{\theta}_t \to \boldsymbol{\theta}_{t+1}$ is *ideally* made through gradient ascent steps on $p(\boldsymbol{y}|\boldsymbol{x}, f)|_{\boldsymbol{y}=\mathcal{I}(\boldsymbol{x})}$, but we instead use the tractable lower bound with $f^t$. We can perform the same derivation using $f^{NN}$ in place of $f^t$. Assuming $p(\boldsymbol{\theta}|\boldsymbol{x}, f^{NN}) \sim q^t(\boldsymbol{\theta})$ for sampling, leads to the improved update rule

$$\boldsymbol{\theta}_{t+1} = \boldsymbol{\theta}_t + \lambda \mathbb{E}_{\boldsymbol{\theta} \sim q^t(\boldsymbol{\theta})} \left[ \nabla_{\boldsymbol{\theta}} \log p(\mathcal{I}(\boldsymbol{x}) | \mathcal{N}(f_{\boldsymbol{\theta}}^{NN}(\boldsymbol{x}), \Sigma)) \right]. \tag{52}$$

Similarly, the negative precision $-\mathbf{H}_{t+1}$ is *ideally* set to be the hessian of the log probability $p(\boldsymbol{\theta}|\boldsymbol{x}, f)$. but instead, we use the tractable lower bound with $f^t$. Here we cannot perform the same derivation, since Theorem 1 does not hold anymore. Instead, we can rely on the estimate Eq. (42) to improve the term

$$\nabla_{\boldsymbol{\theta}}^2 \log p(\boldsymbol{\theta}|\boldsymbol{x}, f^t)\Big|_{\boldsymbol{\theta}=\boldsymbol{\theta}_{t+1}} = -\mathbf{H}_t \qquad \longrightarrow \qquad \nabla_{\boldsymbol{\theta}}^2 \log p(\boldsymbol{\theta}|\boldsymbol{x}, f^{NN})\Big|_{\boldsymbol{\theta}=\boldsymbol{\theta}_{t+1}} \approx -(1-\alpha)\mathbf{H}_t,$$

and this leads to the improved update rule

$$\mathbf{H}_{t+1} = (1-\alpha)\mathbf{H}_t + J_{\boldsymbol{\theta}}f_\cdot^t(\boldsymbol{x})^\top \Sigma^{-1} J_{\boldsymbol{\theta}}f_\cdot^t(\boldsymbol{x}). \tag{53}$$

## E   Fast Hessian

We are interested in computing the hessian of a loss function. For this purpose, the Jacobian of the NN w.r.t. parameters plays a crucial role. In this section we develop a better understanding of this object, we derive the backpropagation (also used by the BackPack library (Dangel et al., 2020)) and finally, we explain our approximated backpropagation that allows linear scaling.

### E.1   Jacobian of a Neural Network

Let us first define some terminology that we will need for chain rule derivations. A NN is a composition of $l$ functions $f := f_{L_l} \circ f_{L_{l-1}} \circ \ldots \circ f_{L_2} \circ f_{L_1}$:

$$x_0 \xrightarrow{f_{L_1}} x_1 \longrightarrow \quad \ldots \quad \longrightarrow x_{i-1} \xrightarrow{f_{L_i}} x_i \longrightarrow \quad \ldots \quad \longrightarrow x_{l-1} \xrightarrow{f_{L_l}} x_l$$

where there are parametric and non-parametric function $f_{L_i}$. We here highlight two common parametric functions and a common non-parametric function.

**Parametric function** such as a linear layer

$$x_i = f_{L_i}(x_{i-1}) = \phi_{L_i} x_{i-1} \quad \text{where } \phi_{L_i} \in \mathfrak{M}(|x_i|, |x_{i-1}|)$$

or convolution

$$x_i = f_{L_i}(x_{i-1}) = \mathtt{conv}_{\text{feat}=\phi_{L_i}}(x_{i-1}) \quad \text{where } \phi_{L_i} \in \mathfrak{M}(\text{in channel, out channel, feat height, feat width})$$

**Non-parametric** such as activation functions $L_i = \mathtt{tanh}, \mathtt{ReLU} \ldots$

$$x_i = f_{L_i}(x_{i-1}) \quad \text{where } |x_i| = |x_{i-1}|$$

What is conventionally called *layer* is actually a composition of two functions: a linear function and an activation function. For sake of clarity in our derivation, we do not adopt this convention and we use the word "layer" to indicate the singular "function" component of the NN.

Let us now consider a NN with $w$ parametric layers and $l - w$ activation layers

$$x_l = f_\phi(x_0) = f_{L_l} \circ \cdots \circ f_{L_1}(x_0) \qquad \text{where } \phi = (\phi_1, \ldots, \phi_w),$$

and define the bijection

$$\mathcal{W}: \quad \{1, \ldots, w\} \quad \longrightarrow \quad \{i| \text{ s.t. } L_i \text{ is parametric layer}\} \quad \subseteq \{1, \ldots, l\}$$
$$p \quad \longmapsto \quad i \text{ s.t. } L_i \text{ has parameters } \phi_p$$

from the subset of parametric layers to the corresponding index in $\phi = (\phi_1, \ldots, \phi_w)$.

The Jacobian w.r.t. the parameters $J_\phi f_\phi(x_0) \in \mathfrak{M}(|x_l|, |\phi_1| + \cdots + |\phi_w|)$ is a matrix with number-of-output $|x_l|$ rows and number-of-parameters $|\phi|$ columns. For reference, just storing this matrix can exceed memory limits even with the smallest autoencoder working on MNIST.

There are two ways of looking at this matrix: (1) **row by row**, that is output by output or (2) **block-of-columns by block-of-columns**, that is layer by layer.

## E.2 Jacobian per output

Each row of the Jacobian corresponds to the gradient w.r.t. the parameters of an element of the output

$$J_\phi f_\phi(x_0) = \begin{pmatrix} \nabla_\phi [f_\phi(x_0)]_1 \\ \vdots \\ \nabla_\phi [f_\phi(x_0)]_{|x_l|} \end{pmatrix}$$

This can be computed by defining $|x_l|$ loss functions

$$loss_k(x_l) := [x_l]_k \quad \text{for } k = 1, \dots, |x_l|$$

and backpropagating each of those to obtain one line at a time. The disadvantage of this formulation is that we cannot reuse computation for one loss function to improve the computation of other loss functions (they are *independent*). Moreover, we need to store all these rows at the same time in order to compute $J^\top J$, which is computationally impractical.

## E.3 Jacobian per layer

Each column of the Jacobian is the derivative of the output vector w.r.t. a single parameter. We can then group the parameters (i.e. columns) layer by layer

$$J_\phi f_\phi(x_0) = \begin{pmatrix} J_{\phi_1} f_\phi(x_0) & \cdots & J_{\phi_w} f_\phi(x_0) \end{pmatrix}$$

where $J_{\phi_p} f_\phi(x_0) \in \mathfrak{M}(|x_l|, |\phi_p|)$. Let us focus on the computation of $J_{\phi_p} f_\phi(x_0)$ for a fixed layer $p = 1, \dots, w$. First notice that the parameters $\phi_p$ in $f_\phi = f_{L_l} \circ \dots \circ f_{L_1}$ only appear in $f_{L_{\mathcal{W}(p)}}$ and so

$$\frac{\partial f_{L_i}}{\partial \phi_p}(x_{i-1}) = 0 \text{ if } i \neq \mathcal{W}(p).$$

**Chain rule** (informal)

$$\frac{\partial f_\phi(x_0)}{\partial \phi_p} = \frac{\partial x_l}{\partial \phi_p} =$$

$$= \frac{\partial x_l}{\partial x_{l-1}} \frac{\partial x_{l-1}}{\partial \phi_p} =$$

$$= \underbrace{\frac{\partial x_l}{\partial x_{l-1}}}_{\substack{\text{layer } l \\ \text{w.r.t. input}}} \underbrace{\frac{\partial x_{l-1}}{\partial x_{l-2}}}_{\substack{\text{layer } l-1 \\ \text{w.r.t. input}}} \cdots \underbrace{\frac{\partial x_{\mathcal{W}(p)+1}}{\partial x_{\mathcal{W}(p)}}}_{\substack{\text{layer } \mathcal{W}(p)+1 \\ \text{w.r.t. input}}} \underbrace{\frac{\partial x_{\mathcal{W}(p)}}{\partial \phi_p}}_{\substack{\text{layer } \mathcal{W}(p) \\ \text{w.r.t. parameters}}}$$

**Chain rule** (formal)

$$J_{\phi_p} f_\phi(x_0) = \left( \prod_{\mathcal{W}(p)+1}^{k=l} J_{x_{k-1}} f_{L_k}(x_{k-1}) \right) J_{\phi_p} f_{L_{\mathcal{W}(p)}}(x_{\mathcal{W}(p)-1}) \tag{54}$$

The intuition for the chain rule is that the Jacobian $J_{\phi_p} f_\phi(x_0)$ is the composition of the Jacobians w.r.t. *input* of subsequent layers times the Jacobian w.r.t. *parameters* of the specific layer. Thus, we can reuse computation for one layer to improve the computation of other layers, specifically the product of Jacobians w.r.t. input. Moreover, we can compute $J_p^\top J_p$ layer by layer without ever storing the full Jacobian.

### E.3.1 Jacobian of a layer w.r.t. to input

The Jacobian of a standard **linear layer** w.r.t. to the input is

$$J_{x_{p-1}} f_{\phi_p}(x_{p-1}) = \phi_p$$

and this remains the same also in the case with a bias. The Jacobian of a **convolutional layer** w.r.t. to the input is

$$J_{x_{p-1}} \text{conv}_{\text{feat}=\phi_p}(x_{p-1}) = \mathcal{M}(\text{conv}_{\text{feat}=\phi_p})$$

The Jacobian of the activation function depends on the specific choice. Recall that, for each layer $i$, $x_i \in \mathbb{R}^{|x_i|}$ is a vector

$$x_i = \left([x_i]_k\right)_{k=1,\ldots,|x_i|}$$

where $[x_i]_k \in \mathbb{R}$ is the value in position $k$ of the vector $x_i$.

If $L_i$ is **tanh**

$$[x_i]_k = [f_{L_i}(x_{i-1})]_k = \text{tanh}([x_{i-1}]_k) = \frac{e^{[x_{i-1}]_k} - e^{-[x_{i-1}]_k}}{e^{[x_{i-1}]_k} + e^{-[x_{i-1}]_k}} \qquad \text{for } k = 1,\ldots,|x_{i-1}|$$

then

$$[J_{x_{i-1}} f_{L_i}(x_{i-1})]_{kj} = \delta_{kj}\left(1 - (\text{tanh}([x_{i-1}]_k))^2\right) = \delta_{kj}\left(1 - [x_i]_k^2\right) \qquad \text{for } k,j = 1,\ldots,|x_{i-1}|.$$

If $L_i$ is **ReLU**

$$[x_i]_k = [f_{L_i}(x_{i-1})]_k = \text{ReLU}([x_{i-1}]_k) = \max(0, [x_{i-1}]_k) \qquad \text{for } k = 1,\ldots,|x_{i-1}|$$

then

$$[J_{x_{i-1}} f_{L_i}(x_{i-1})]_{kj} = \delta_{kj}\left(1 \text{ if } [x_{i-1}]_k > 0 \text{ else } 0\right) \qquad \text{for } k,j = 1,\ldots,|x_{i-1}|.$$

### E.3.2   Jacobian of a layer w.r.t. to parameters

The Jacobian of a standard linear layer w.r.t. to the parameters is

$$J_{\phi_i} f_{\phi_i}(x_{i-1}) = \mathbb{I}_{|x_i|} \otimes x_{i-1} \qquad \in \mathfrak{M}(|x_i|, |\phi_i|)$$

and in the case with bias $b_i \in \mathbb{R}^{x_i}$ the Jacobian is

$$J_{\phi_i, b_i} f_{\phi_i, b_i}(x_{i-1}) = \mathbb{I}_{|x_i|} \otimes [x_{i-1}, 1] \qquad \in \mathfrak{M}(|x_i|, |\phi_i| + |b_i|)$$

The Jacobian of a convolutional layer w.r.t. to the parameters is

$$J_{\phi_i} \text{conv}_{\text{feat}=\phi_i}(x_{i-1}) = J_{\phi_i} \text{conv}^T_{\text{feat}=rev(x_{i-1})}(\phi_i) = \mathcal{M}(\text{conv}_{\text{feat}=rev(x_{i-1})})^T \qquad \in \mathfrak{M}(|x_i|, |\phi_i|)$$

and in the case with bias $b_i \in \mathbb{R}^{o_i}$ the Jacobian is

$$J_{\phi_i} \text{conv}_{\text{feat}=\phi_i, \text{bias}=b_i}(x_{i-1}) = \left(\mathcal{M}(\text{conv}_{\text{feat}=rev(x_{i-1})})^T \big| I\right) \qquad \in \mathfrak{M}(|x_i|, |\phi_i| + |b_i|)$$

### E.4   Hessian of a Neural Network

Consider a function $\mathcal{L}: \mathbb{R}^{|x_l|} \to \mathbb{R}$ from the output of the NN to scalar value. This later will be interpreted as loss or likelihood, but for now, let us stick to the general case.

We are interested in the hessian of this scalar value w.r.t. the parameters of the NN

$$\nabla^2_\phi\left(\mathcal{L}(f_\phi(x_0))\right) \in \mathfrak{M}\left(\sum_{p=1}^w |\phi_i|, \sum_{p=1}^w |\phi_i|\right).$$

Similarly to the previous section, it is convenient to see this matrix as block matrices, separated layer-wise

$$\nabla^2_\phi \mathcal{L}(f_\phi(x_0)) = \begin{pmatrix} \nabla^2_{\phi_1} \mathcal{L}(f_\phi(x_0)) & \frac{\partial^2}{\partial\phi_1\partial\phi_2}\mathcal{L}(f_\phi(x_0)) & \cdots & \frac{\partial^2}{\partial\phi_1\partial\phi_w}\mathcal{L}(f_\phi(x_0)) \\ \frac{\partial^2}{\partial\phi_2\partial\phi_1}\mathcal{L}(f_\phi(x_0)) & \nabla^2_{\phi_2}\mathcal{L}(f_\phi(x_0)) & & \\ \vdots & & \ddots & \\ \frac{\partial^2}{\partial\phi_w\partial\phi_1}\mathcal{L}(f_\phi(x_0)) & & & \nabla^2_{\phi_w}\mathcal{L}(f_\phi(x_0)) \end{pmatrix}$$

The first common assumption is to consider layers to be independent of each other, i.e.

$$\frac{\partial^2}{\partial \phi_i \partial \phi_j} \mathcal{L}(f_\phi(x_0)) = 0 \qquad \forall i \neq j$$

Let us now fix a layer $p = 1, \ldots, w$ and focus on a single diagonal block.

$$\nabla^2_{\phi_p} \mathcal{L}(f_\phi(x_0)) \in \mathfrak{M}(|\phi_p|, |\phi_p|).$$

According to the chain rule

$$\nabla^2_{\phi_p} \mathcal{L}(f_\phi(x_0)) = \underbrace{J_{\phi_p} f_\phi(x_0)^T \cdot \nabla^2_{x_l} \mathcal{L}(x_l) \cdot J_{\phi_p} f_\phi(x_0)}_{=:G(\phi)} + \sum_{i=1}^{|x_l|} [\nabla_{x_l} \mathcal{L}(x_l)]_i \cdot \nabla^2_{\phi_p} [f_\phi(x_0)]_i \quad (55)$$

The second term of the RHS is equal to 0 if the model perfectly fits the dataset, $\nabla_{x_l} \mathcal{L}(x_l) = 0$, OR if $f$ is linear in the parameters, $H_{\phi_p} [f_\phi(x_0)]_i = 0$.

The first term of the RHS of Eq. 55, $G(\phi)$, is in literature referred to as Generalized Gauss-Newton (GGN) matrix. It can be computed efficiently thanks to the view of the Jacobian as layer by layer. Using equation (54), the expression for the approximated hessian w.r.t. to $\phi_p$ is then

$$G(\phi) = J_{\phi_p} f_\phi(x_0)^T \cdot H_{x_l} \mathcal{L}(x_l) \cdot J_{\phi_p} f_\phi(x_0) =$$

$$= J_{\phi_p} f_{L_{\mathcal{W}(p)}}^T \left( \prod_{k=\mathcal{W}(p)+1}^{l} J_{x_k} f_{L_k}^T \right) H_{x_l} \mathcal{L}(x_l) \left( \prod_{\mathcal{W}(p)+1}^{k=l} J_{x_k} f_{L_k} \right) J_{\phi_p} f_{L_{\mathcal{W}(p)}}$$

and from this, we can build an efficient backpropagation algorithm.

---

**Algorithm 1** Algorithm for $J_\phi f^T \cdot \nabla^2 \mathcal{L} \cdot J_\phi f$

---

$M = \nabla^2_{x_l} \mathcal{L}(x_l)$
**for** $k = l, l-1, \ldots, 1$ **do**
    **if** $L_k$ is parametric with $\phi_p$ (i.e. $k = \mathcal{W}(p)$): **then**
        $H_p = J_{\phi_p} f_{L_k}^\top \cdot M \cdot J_{\phi_p} f_{L_k}$
    **end if**
    $M = J_{x_k} f_{L_k}^\top \cdot M \cdot J_{x_k} f_{L_k}$
**end for**
**return** $(H_1, \ldots, H_w)$

---

As it is written, each $H_p$ is a matrix $|\phi_p| \times |\phi_p|$ so we technically obtain the hessian in a block-diagonal form

$$\begin{pmatrix} H_1 & & 0 \\ & \ddots & \\ 0 & & H_w \end{pmatrix}$$

if we are interested in the diagonal only, we can construct that by concatenation of the diagonals for each $H_p$.

### E.4.1 Fast approximated version of the Algorithm

The idea is to backpropagate only the diagonal of the matrix $M$, neglecting all the non-diagonal elements. In a single backpropagation step, we have

$$M' = J_{x_p} f_{L_p}^\top \cdot M \cdot J_{x_p} f_{L_p}$$

In order to backpropagate the diagonal only, we need to use the operator

$$\text{diag}(M) \mapsto \text{diag}(M')$$

For linear layers and activation functions, this operator is trivial. For the convolutional layer it turns out that this operator is itself a convolution

$$\text{diag}(M') = \text{conv}_{\text{feat}=\phi_p^{(2)}} (\text{diag}(M))$$

where the kernel tensor $\phi_p^{(2)}$ is the pointwise square of the kernel tensor $\phi_p$.

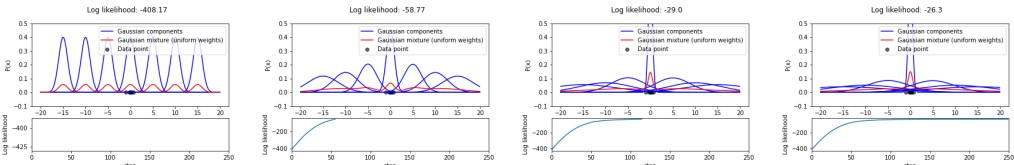

Figure 1: Snapshots of optimization of a mixture of Gaussians with fixed means and component weights (i.e. only variances are learned). We observe that variances of components far away from data increase in order to push more probability mass to the region where data resides. The Gaussian VAE should exhibit the same behavior in order to maximize data likelihood, but in practice, it does not.

### E.5 Hessian of a Reconstruction Loss

In the previous Algorithm, the backpropagated quantity is initialized as the hessian of the loss w.r.t. the output of the NN $\nabla^2_{x_l}\mathcal{L}(x_l)$, or, in an equivalent but more compact notation $\nabla^2_f\mathcal{L}(f)$. The value of this hessian clearly depends on the specific choice of loss function $\mathcal{L}$.

The most common choice of the likelihood for regression is the Gaussian distribution, while for classification it is the Bernoulli distribution. The Gaussian log-likelihood is

$$\mathcal{L}(f) := \log p(x|\mu = f_\theta(x), \sigma^2 = \sigma_d^2) = -\frac{1}{2\sigma_d^2}\|x - f_\theta(x)\|^2 - \log(\sqrt{2\pi}\sigma_d) \tag{56}$$

and its hessian is identity scaled with $\sigma_d$

$$\nabla^2_f \log p(x|\mu = f_\theta(x), \sigma^2 = \sigma_d^2) = -(\sigma_d)^{-2}\mathbb{I} \tag{57}$$

The Bernoulli log-likelihood is

$$\mathcal{L}(f) := \log p(c|f_\theta(x)) = \log[\texttt{softmax}(f_\theta(x))]_c = [f_\theta(x)]_c - \log\left(\sum_i e^{[f_\theta(x)]_i}\right) \tag{58}$$

and its hessian can be written in terms of the vector $\pi = \texttt{softmax}(f_\theta(x))$ of predicted probabilities

$$\nabla^2_f \log p(c|f_\theta(x)) = -\nabla^2_f \log\left(\sum_i e^{[f_\theta(x)]_i}\right) = diag(\pi) - \pi\pi^T \tag{59}$$

We highlight that both Hessians are independent on the label, and thus the GGN is equal to the Fisher matrix, this is true every time $p(y|f_\theta(x))$ is an exponential family distribution with natural parameters $f_\theta(x)$. In this paper, we focus on the Gaussian log-likelihood, but we emphasize that the method is not limited to this distribution.

## F  Intuition on optimization of variance in VAEs

One can imagine the Gaussian VAE as an infinite mixture of Gaussians (see e.g. Mattei and Frellsen (2018) for an extensive discussion of this link) where the weights are fixed by the prior on the latent space. To increase the probability of the training data $p(x)$, optimizing the neural network will push the probability mass from regions far away from training data to regions with training data. Thus, the network should learn to have large variance far away from training data, and low variance close to training data.

We illustrate this idea with a toy example (see snapshots in Fig. 1 or animation at `https://frederikwarburg.github.io/gaussian_vae.html`). In this example, we show a mixture of Gaussian with components, where we optimize the variance (and fix the mean and the weights of the mixture components). We see that the variance of the components far away from the training data increases, whereas the variance of the component close to the data decreases. The opening example of the paper demonstrates that the Gaussian VAE does not exhibit this behavior even if this is optimal in terms of data likelihood.