# OpenReview forum: "Laplacian Autoencoders for Learning Stochastic Representations"
_NeurIPS.cc/2022/Conference — NeurIPS 2022 Accept_

### Official Review · Reviewer_4XuV · 2022-07-11

**Rating:** 7
**Confidence:** 4
**Soundness:** 4 excellent
**Presentation:** 4 excellent
**Contribution:** 4 excellent

**Summary:**

The authors propose to use Bayesian autoencoders for unsupervised representation learning, where they approximate the posterior over the parameters using a Laplace approximation (LA). In contrast to previous work, they update the LA during training, for which they devise a Monte Carlo EM iterative update scheme. They show that this approach performs really well on a range of different problems.

**Questions:**

Overall, this is a really nice paper and I don't have much to criticize. Just a few minor comments below.

- l. 24: "VAEs" -> "VAE's"
- Fig. 1 caption: "laplace" -> "Laplace"
- l. 136: The fact that the GGN is the exact Hessian in the linearized model has also been noted by Immer et al. [1]
- l. 170: "sensible" -> "sensitive"


[1] https://proceedings.mlr.press/v130/immer21a.html

**Limitations:**

The authors have addressed all limitations and ethical concerns.

**Strengths And Weaknesses:**

Strengths:

- The paper is well written and the contributions are clear.
- The Monte Carlo EM update seems like an elegant solution for training with Laplace approximations.
- The empirical performance of the proposed method is impressive.

Weaknesses:

- A few typos (see below)

---

> ### Author Response · Authors · 2022-08-02
> **Reply to reviewer 4XuV**
>
> We thank the reviewer for the positive remarks and comments. We will clarify in the final version that Immer et al. also note that GGN is the exact Hessian for the linearized model. We will also correct the typos highlighted by the reviewer.

---

### Official Review · Reviewer_meVs · 2022-07-11

**Rating:** 6
**Confidence:** 3
**Soundness:** 3 good
**Presentation:** 2 fair
**Contribution:** 3 good

**Summary:**

This paper introduces a novel Bayesian autoencoder. By applying Monte Carlo EM with a variational distribution with Laplacian approximation, the proposed model become tractable and can perform iterative updates. To make the proposed method scalable to large datasets, the authors also propose to approximate the Hessian matrix to make the running time scales linearly with the input size. Experiment results show that the proposed method perform well on tasks like OOD detection and data imputation.

**Questions:**

1. In Table 2, why the exact method (with post-hoc) performs worse than the approximation method (in both the log-likelihood and MSE)? Are the metric diffs big enough to say that the approximation method performs better here than the exact method? If so, it seems weird and it will be good if the authors can check the implementations. If not, it will be better if the authors can show the confidence intervals of these metrics.

2. The authors proposed to apply diagonal approximation for the Hessian. This is definitely good for large scale training. However, a diagonal approximation is probably an over approximation. Did you try some idea with (potentially) tighter approximations?

**Limitations:**

This paper discussed about the limitations of their work.

**Strengths And Weaknesses:**

Strengths:

1. The idea of applying LA and Monte Carlo EM is reasonable. Through these applications, the neural networks can be sampled and the updates for the parameters become a normal set of updates for posterior computations, which is tractable.

2. The proposed updates can ensure the Hessian part is always negative definite, which makes the process smooth.

3. The Hessian diagonal approximation makes sure that the proposed method can scale up to higher dimensionality.

Weaknesses:

1. This paper is hard to follow and many details are missing or hard to find. For example, the second paragraph of Section 1.1 is unclear. It also has some confusing presentations (e.g. the first sentence says that $\sigma^2(z)$ should be as large as possible to make $p(x)$ large, it seems like this is not the case) or some typos (e.g. Eq 1 is a lower bound on $\log p(x)$, not $p(x)$ as shown there). Another example is Figure 1, where no detailed introduction about the definition of this figure was shown in the caption. It will be better if the presentation can be improved.

---

> ### Author Response · Authors · 2022-08-02
> **Reply to reviewer meVs**
>
> We appreciate the reviewer for highlighting that parts of the presentation can be further improved. We have addressed all the reviewer's comments and updated the manuscript based on these comments.

---

> > ### Author Response · Authors · 2022-08-02
> > **Future work on tighter approximation?**
> >
> > We agree that the approximate diagonal approximation of the Hessian is a crude approximation. We thought much about tighter approximations but did not come up with any that scales linearly with the image resolution, which is a requirement to scale to large images and for practical applications. In practice, we found that the approximate diagonal Hessian works well, but we agree that tighter approximations are of interest.
> >
> > We will add these considerations to the section of future work.

---

> > ### Author Response · Authors · 2022-08-02
> > **Standard deviation for results in Table 2**
> >
> > We have updated the table with confidence intervals (one standard deviation). Since obtaining the normal distribution over the parameters by fitting the Hessian in the post-hoc approach is deterministic, we train $5$ networks and fit the post-hoc Laplace approximation to each of them. To further validate that our implementation is correct, we compute the post-hoc both using our and the Laplace Redux~(https://github.com/AlexImmer/Laplace) implementation. For the online methods, we train from scratch $5$ times. Table 1 shows the revised table.
> >
> > | Hessian     | $\log p(x) \uparrow$   | MSE $\downarrow$   |
> > |-------------|------------------------|--------------------|
> > | KFAC$^{**}$ | $-9683.91 \pm 2454.95$ | $121.55 \pm 24.49$ |
> > | Exact$^{**}$ | $-283.34 \pm 88.62$    | $27.11 \pm 0.86$   |
> > | Exact       | $-282.57 \pm 88.70$    | $27.14 \pm 0.95 $  |
> > | Approx      | $-232.97 \pm 65.48$    | $26.57 \pm 0.64$   |
> > | Exact$^{*}$  | $-25.77 \pm 0.20$      | $25.72 \pm 0.21$   |
> > | Approx$^*$  | $-25.88 \pm 0.38$      | $25.83 \pm 0.38$   |
> > **Table 1:** _Online training (indicated by $^*$) outperforms post-hoc LA. $^{**}$ are computed with Laplace Redux implementation. The approximate diagonal always has a similar performance to the exact diagonal._
> >
> > First, Table 1 shows no difference between our implementation of the exact Hessian (Exact) and the Laplace Redux implementation (Exact$^{**}$) (https://github.com/AlexImmer/Laplace).
> >
> > Second, the results show that there is not a significant difference between the approximate and exact diagonal Hessian in neither the post-hoc nor online method, however, the proposed online training significantly improves the likelihood compared to the post-hoc approach.
> >
> > We further emphasize that the exact diagonal Hessian (which is commonly used for the Laplace approximation) is also an approximation of the full Hessian and that the Laplace approximation is an approximation of the true posterior distribution. This means that common implementations of the Laplace method are already making heavy approximations, thus, it is not surprising that further approximating the diagonal Hessian performs on par with the exact diagonal Hessian. We will elaborate further on this intuition and include the revised table in the paper. We hope that the reviewer will consider upgrading their score based on these improvements to the manuscript.

---

> > ### Author Response · Authors · 2022-08-02
> > **Improved clarity of the presentation**
> >
> > **Intuition on optimization of variance:** We believe that the confusion of the sentence:
> >
> > _If we momentarily assume that $p(x | z) = \mathcal{N}(x | \mu(z), \sigma^2(z))$, we see that optimally $\sigma^2(z)$ should be as large as possible away from training data in order to increase $p(x)$ on the training data._
> >
> > stems from the conditioning on the **training data**. One can imagine the Gaussian VAE as an infinite mixture of Gaussians (see e.g. Mattei \& Frellsen [1] for an extensive discussion of this link) where the weights are fixed by the prior on the latent space. To increase the probability of the training data $p(x)$, optimizing the neural network will push the probability mass from regions far away from training data to regions with training data.
> > Thus, the network should learn to have large variance far away from training data, and low variance close to training data.
> >
> > We illustrate this idea with a toy example (see https://pasteboard.co/9oyjUby9HR6i.gif). In this example, we show a mixture of Gaussian with $5$ components, where we optimize the variance (and fix the mean and the weights of the mixture components). We see that the variance of the components far away from the training data increases, whereas the variance of the component close to the data decreases. We hope the reviewer finds this explanation satisfactory. We will clarify these points in the final manuscript.
> >
> > [1] Mattei P-A, Frellsen J (2018) Leveraging the Exact Likelihood of Deep Latent Variable Models. Advances in Neural Information Processing Systems 31 (NeurIPS 2018).
> >
> >
> > **Typo in equation 1:**
> > We thank the reviewer for spotting the typo in Eq. 1. We will change $p(x)$ to $\log p(x)$ in the updated version. We highlight that this change does not affect the points made in the section; namely that (1) the lower bound of the VAE should result in large variance far away from the training data and (2) that this does not happen in practice because the neural network $\sigma^2$ is never trained in regions far away from the training data.
> >
> >
> > **Improved caption of figure 1**
> > We agree with reviewers 1 and 2 that the caption of Fig. 1 can be improved to better explain the details of the figure. We have made the following changes to clarify the details:
> >
> > _Figure 1: 2D latent representation of mnist overlaid a heatmap that describes the decoder uncertainty. Yellow/blue indicates low/high variance of the reconstructions. The mean and variance of a reconstruction are shown next to the latent space. (a) The VAE learns to estimate high variance for low-level image features such as edges but fails at extrapolating uncertainties away from training data. (b) Applying post-hoc Laplace to the AE setup shows much better extrapolating capabilities, but fails in estimating calibrated uncertainties in output space. (c) Our online, sampling-based optimization of a Laplacian autoencoder (LAE) gives well-behaved uncertainties in both latent and output space._
> >
> > We hope that the reviewer will agree that these changes improve the clarity of the section and the paper. If you find other sections/paragraphs hard to follow, we'd love the associated feedback in order to improve the manuscript.

---

### Official Review · Reviewer_vo1U · 2022-07-12

**Rating:** 7
**Confidence:** 3
**Soundness:** 4 excellent
**Presentation:** 4 excellent
**Contribution:** 3 good

**Summary:**

The authors propose a novel framework for learning stochastic representations using a Bayesian autoencoder called Laplacian Autoencoders. They derive a novel training objective to train this Bayesian autoencoder and propose a novel approximation for the fast computation of the Hessian that makes their approach viable on high-dimensional datasets. The experiments section demonstrates the advantage of this approach over previous ones (notably over a post-hoc analysis of a pretrained autoencoder) on tasks like out-of-distribution detection, missing data imputation or semi-supervised learning.

**Questions:**

Here are some suggestions that could help:
I would suggest to find a way to better detail the derivation of p. 3: conditionning on f is hard to grasp and I had troubles following all the details. Some ideas might need to be justified as they can appear surprising for a non expert of the domain (e.g. using the Hessian of the lower bound instead of the true term; the introduction of the $\alpha$ parameter).

The caption of Fig. 1 could precise what is shown.
There are some typos: many concerning missing s on the third person (e.g. ll. 58, 128, 205, 212),  what is the difference between $f^{(t)}$, $f^{(t)}_{\theta_t}$, $f_{\theta_t}$?

**Limitations:**

The authors adequately addressed the limitations and potential negative societal impact of their work.

**Strengths And Weaknesses:**

This paper is very well-written: both the writing and the presentation are precise and clear. It is well illustrated and the experiments show the relevance of this approach on a wide variety of tasks. The problem is properly introduced via a well-chosen example and the related works section properly discusses the  relevant literature.
I still had some troubles with pp. 3 and 4 due to some notations issues or quick derivations.
The problem tackled will interest many researchers in the community and this article proposes both a clear and principled derivation of the objective together with practical considerations.

---

> ### Author Response · Authors · 2022-08-02
> **Reply to reviewer vo1U**
>
> We thank the reviewer for the detailed considerations and questions. In the following we address each of the questions; first with a short summary answer, followed by a longer and more detailed answer.
>
> To begin with, we wish to accentuate our experimental contribution. As the reviewer correctly points out, our online LAE outperforms the post-hoc LAE across multiple tasks (out-of-distribution detection, missing data imputation, and semi-supervised learning). However, we highlight that the post-hoc LAE is also a contribution of the paper and that this method in turn outperforms a standard VAE. We emphasize that the VAE is the main baseline and that both our post-hoc and online LAE naturally handles many of the things people usually struggle with in VAEs such as OOD detection, missing data imputation, and semi-supervised learning. On top of that, the LAE is very easy to train and robust to hyper-parameter choices.

---

> > ### Author Response · Authors · 2022-08-02
> > **Figure 1 and typos**
> >
> > We agree with reviewers 1 and 2 that the caption of Fig. 1 can be improved to better explain the details of the figure. We have made the following changes to clarify the details:
> >
> >
> > _Figure 1: 2D latent representation of mnist overlaid a heatmap that describes the decoder uncertainty. Yellow/blue indicates low/high variance of the reconstructions. The mean and variance of a reconstruction are shown next to the latent space. (a) The VAE learns to estimate high variance for low-level image features such as edges but fails at extrapolating uncertainties away from training data. (b) Applying post-hoc Laplace to the AE setup shows much better extrapolating capabilities, but fails in estimating calibrated uncertainties in output space. (c) Our online, sampling-based optimization of a Laplacian autoencoder (LAE) gives well-behaved uncertainties in both latent and output space._
> >
> >
> > We further went through the paper for extra proofreading and fixed third person s's as well as other typos. We hope that the reviewer agrees that these changes improve the clarity of the paper, and hope the reviewer will consider upgrading their score.

---

> > ### Author Response · Authors · 2022-08-02
> > **The introduction of the parameter**
> >
> > Again, we thank the reviewer for highlighting that the introduction of $\alpha$ can appear surprising for non-experts. The introduction of this parameter stems from both practical and theoretical reasons. On one hand, it can be viewed as a geometric running average that is useful for smoothing out results computed on a minibatch instead of on the full training set. This is similar to momentum-like training procedures.
> >
> > Secondly, it allows for non-monotonically-increasing precision. Note that we revisit data, meaning that during training the same datapoint is revisited several times and for every visit, we update the precision matrix. Thus, this forgetting induced by $\alpha$ is a wanted behavior in order to avoid infinite precision in the limit of training time.
> >
> > We further investigated why the precision was actually non-monotonically increasing and realized that an approximation is made from one neural network linearization to the next one. This difference leads to different information gain when the datapoint is revisited for a second time, but with some different neural network linearization. In a nutshell: \emph{we can never condition on the dataset, we condition only on the image of the dataset through the neural network with some specific parameter}. Intuitively, one can think of the precision matrix changes as a consequences of (1) information gain from the new network evaluation, i.e. addition of a positive definite matrix, and (2) information loss from the old network linearization being forgotten, i.e. subtraction of a negative definite matrix.
> > Thus the parameter $\alpha$ is introduced to deal with an unknown quantity regarding the difference in the shape of the probabilities conditioned to $f$ or $f^{(t)}$. This quantity is the Hessian of the log-likelihood of $\theta$ conditioned to the data. We never want to overestimate the precision, so we are interested in a lower bound for this unknown quantity. Setting the parameter $\alpha$ to be in the same order of magnitude of the learning rate, empirically showed good results. This is some piece of evidence that we deal with a reasonable approximate precision lower bound.
> >
> > We will include these considerations in the main text when we introduce $\alpha$.

---

> > ### Author Response · Authors · 2022-08-02
> > **Using Hessian of the lower bound instead of true term**
> >
> > We thank the reviewer for highlighting that this might appear surprising to non-experts. The choice of using the lower bound for the Hessian is driven mainly by reasons of consistency. In fact, this is the same objective we maximize and on which we make gradient steps. Secondly, similarly to the first order derivative, it is simply more straightforward to compute by approximating the expectation with Monte Carlo sampling. We found this to work very well, as shown in the experimental section and did not explore these alternatives. We believe these alternatives could also work well, but they will probably lead to a more complicated and less consistent model. We will clarify the choice in the updated manuscript.

---

> > ### Author Response · Authors · 2022-08-02
> > **Conditioning of $f$**
> >
> >
> > **Short:** As stated above, $f$ and $f^{(t)}$ express two different quantities, thus we need to differentiate between the distribution induced by $f$ from the one induced by the linearized $f^{(t)}$. An alternative notation to the one used in the current manuscript is to use superscripts such as $p^f(\theta,x,x_ {\text{rec}})$, however in our opinion this leads to less readable equations.
> >
> > **Long:** The Bayesian setting deals with the joint distribution $p(x_ {\text{rec}},x,\theta)$ and all its marginals and conditionals.
> > Classically we seek an expression of $p(\theta|x)$ that is, in some sense, optimal. Figure 3 in the paper shows two examples of $p(x_ {\text{rec}},\theta|x)$. In these two examples, all assumptions are the same, the only difference is the choice of the network architecture. From the figure, we observe that the distribution deduced depends on the choice of network architecture. Thus, referring to the joint distribution without further specifying if we are conditioning on $f$ or the linearized $f^{(t)}$ is improper.
> >
> > One option for specifying the choice of network architecture is to use a superscript
> >
> > \begin{equation}
> >     p^f(\theta,x,x_{\text{rec}})
> >     \not=
> >     p^{f'}(\theta,x,x_{\text{rec}})
> >     \qquad
> >     \text{if }\quad f\not=f'
> > \end{equation}
> >
> >
> > maybe together with the shorthand $p^t(\theta,x,x_\text{rec})$ in place of $p^{f^{(t)}}(\theta,x,x_ \text{rec})$. However, we found this superscript notation to be less readable and less fit for viewing the problem in a more generalizable setting.
> >
> > Instead, we choose to exploit that the dependence on the network architecture stems from the definition of the marginal $p^f(x_ {\text{rec}}|x,\theta) \sim \mathcal{N}(x_ {\text{rec}}|\mu=f_ \theta(x),\Sigma=\mathbb{I})$ that the joint $p^f(\theta,x,x_ \text{rec})$ must satisfy. This led us to the notation used in the paper; namely, for any
> > measure on the operator space $\mathbb{F}:\Theta \rightarrow \left(\mathbb{X}\rightarrow\mathbb{Y}\right)$ we assume the previous joint $p^f$ on $\Theta\times \mathbb{X}\times \mathbb{Y}$ to be the marginals
> >
> > \begin{equation}
> >     p(\theta,x,x_ {\text{rec}}|f)
> >     =
> >     p^f(\theta,x,x_ {\text{rec}})
> > \end{equation}
> >
> > of an unknown joint distribution $p(\theta,x,x_ {\text{rec}},f)$ on $\Theta\times\mathbb{X}\times \mathbb{Y}\times\mathbb{F}$.
> > We highlight that this notation is more expressive while maintaining the same clearness.
> >
> > This more expressive and flexible notation might be useful for extensions in future works. For example, the notation allows for modeling stochastic properties on $\mathbb{F}$. Notice that the specific measure choice on $\mathbb{F}$ is directly linked to some continuity of the marginal with regards to the architecture, that one can enforce or exploit. More specifically, by assuming meaningful measure and a prior on $\mathbb{F}$ one can, in principle, perform full Bayesian model architecture selection. This is appealing but requires dealing with functional space and non-infinite-dimensional Lebesgue measure. An interesting approach (we partly share notation with) is the paper ``Functional Variational Bayesian Neural Network'' (Sun et al. 2019) which uses finite subsets to deal with the functional space measure.
> >
> > We thank the reviewer for highlighting that conditioning on $f$ can be hard to grasp. We hope the above-listed considerations clarify the chosen notation and the conditioning on f. We will update the manuscript to better explain the conditioning on $f$.

---

> > ### Author Response · Authors · 2022-08-02
> > **Difference between $f^{(t)}, f^{(t)}_{\theta_t}$,$f_{\theta_t}$**
> >
> > **Short:** The symbols $f$ and $f^{(t)}$ stands for neural network architectures; respectively the neural network and its linearization. The symbols $f_{\theta_t}$ and $f^{(t)}_{\theta_t}$ stands for the architectures equipped with some specific parameter $\theta_t$. We introduce this notation to make a clear distinction between the function spanned by the neural network with given weights, and the weights themselves (i.e. function space vs weight space).
> >
> >
> > **Long:** Neural networks are usually referred to as functions from some input space $\mathbb{X}$ to some output space $\mathbb{Y}$ for any given parameter. In order to make a clear distinction between the function spanned by the neural network with given weights, and the weights themselves (i.e. function space $\mathbb{F}$ vs weight space $\Theta$) we need a bit more rigorous notation.
> >
> > Both $f$ and $f^{(t)}$ are operators in $\mathbb{F}:\Theta \rightarrow \left(\mathbb{X}\rightarrow\mathbb{Y}\right)$ that maps some parameter $\theta\in\Theta$ to some function $f_\theta:\mathbb{X}\rightarrow \mathbb{Y}$. The subscript $\theta$ notation is actually a shorthand for equivalent but heavier notations like $f(\theta):\mathbb{X}\rightarrow \mathbb{Y}$ with $f(\theta)(x)\in\mathbb{Y}$ or $f:\Theta\times\mathbb{X}\rightarrow \mathbb{Y}$ with $f(\theta,x)\in\mathbb{Y}$. Despite being operators in the same spaces, $f$ and $f^{(t)}$ are actually different, meaning that they assume different $y$ values for the same $x$ input (this is true every time that - as it always is with realistic network architectures - $f$ is not linear in $\Theta$).
> >
> > However, $f^{(t)}$ is defined by: _the linearization of $f$ around $\theta^t$_, for some fixed $\theta^t\in\Theta$ from which the superscript $(t)$ in $f^{(t)}$ comes from
> >
> > $$f^{(t)}_ \theta(x) := f_{\theta_t}(x) + J_\theta f_{\theta_t}(x) (\theta-\theta_t)\qquad \forall \theta\in\Theta,x\in X.$$
> >
> > This definition implies some equalities that are useful to keep in mind, like that
> >
> > $$
> >     f^{(t)}_ {\theta_t}(x)
> >     =
> >     f_{\theta_t}(x)
> >     \qquad
> >     \forall x\in X
> > $$
> >
> > since the term $\theta-\theta_t$ is 0 when evaluated in $\theta=\theta_t$. Or, for example, that the Jacobian of $f^{(t)}$ evaluated everywhere is equal to the Jacobian of $f$ evaluated in $\theta_t$
> >
> > $$
> >     J_ \theta f^{(t)}_ {\theta'}(x)
> >     =
> >     J_ \theta f_ {\theta_t}(x)
> >     \qquad
> >     \forall \theta'\in\Theta,x\in X
> > $$
> >
> > and thus it is constant in $\theta$
> >
> > $$
> >     J_ \theta f^{(t)}_ \theta(x)
> >     =
> >     J_ \theta f^{(t)}_ {\theta'}(x)
> >     \qquad
> >     \forall \theta,\theta'\in\Theta,x\in X.
> > $$
> >
> > These identities might lead to confusion but are opted to simplify notation. For example, we use the lighter, but slightly ``sloppier'' notation $J_ \theta f^{(t)}_ \theta(x)$ (such as in Eq (11) in the paper) without further specifying, which specific $\theta$ we evaluate the Jacobian in (being it constant, it does not matter). We opt for this notation to make the equations more comprehensible.
> >
> > We thank the reviewer for raising this point and will update the manuscript and appendix to clarify the difference between the neural network $f$, the linearized neural network $f^{(t)}$, the neural network $f_{\theta_t}$ equipped with some parameters $\theta_t$, and the linearized neural network $f^{(t)}_{\theta_t}$ equipped with some parameters $\theta_t$.

---

### Author Response · Authors · 2022-08-02
**Overall comment**

Dear area chairs and dear reviewers,

We would like to thank you all for your efforts in handling our submission and for providing such excellent feedback on our work. Your comments have been very valuable to improve the quality of our manuscript and our work.

We have addressed all your comments in this rebuttal, and have made a thorough revision of our submission based on them.

We believe that there has been a general comprehension of our work, particularly of the main contributions and novelty. It seems that all the reviewers agree that the work is novel, the solution is elegant, the empirical results are impressive, and the work will interest many researchers in the community.

Below we have answered every question and comment included in the reviews. We are particularly interested in improving the paper as much as possible with the help of the reviewers, and for that reason, we add extra details to the answers.

---

### Meta-Review · Area_Chair_ujES · 2022-08-27

**Recommendation:** Accept
**Confidence:** Certain

**Metareview:**

Thanks to the authors for this submission.  The reviewers were all quite positive — one reviewer noted that the “author response was really thorough and insightful”.  The reviewer-author discussion appeared to be fruitful, and new experimental results were presented to address reviewer questions. The reviewers agreed that the presentation was clear, the experiments thorough, and that the problem addressed will be of broader interest to members of the ML community.

**Award:**

No

---

### Decision · Program_Chairs · 2022-09-14

Accept